# Localized Cultural Knowledge is Conserved and Controllable in Large Language Models

## Abstract

Just as humans display language patterns influenced by their native tongue when speaking new languages, LLMs often default to English-centric responses even when generating in other languages. Nevertheless, we observe that local cultural information persists within the models and can be readily activated for cultural customization. We first demonstrate that explicitly providing cultural context in prompts significantly improves the models' ability to generate culturally localized responses. We term the disparity in model performance with versus without explicit cultural context the *explicit–implicit localization gap*, indicating that while cultural knowledge exists within LLMs, it may not naturally surface in multilingual interactions if cultural context is not explicitly provided. Despite the explicit prompting benefit, however, the answers reduce in diversity and tend toward stereotypes. Second, we identify an explicit cultural customization vector, conserved across all non-English languages we explore, which enables LLMs to be steered from the synthetic English cultural world-model toward each non-English cultural world. Steered responses retain the diversity of implicit prompting and reduce stereotypes to dramatically improve the potential for customization. We discuss the implications of explicit cultural customization for understanding the conservation of alternative cultural world models within LLMs, and their controllable utility for translation, cultural customization, and the possibility of making the explicit implicit through soft control for expanded LLM function and appeal.

## 1 Introduction

When humans speak a second language, they often display patterns of language use influenced by their native tongue (Dong et al., 2005; Matsuki et al., 2021). A striking example of this phenomenon can be observed among Japanese speakers learning English. When asked about the color of a pumpkin, they frequently respond "green" (Matsuki et al., 2021), reflecting the concept's reality in Japan, where pumpkins are predominantly green, rather than the orange variety common in English-speaking countries. Interestingly, these same speakers, when specifically asked about pumpkin colors in America, will correctly identify them as orange.

Large language models (LLMs) exhibit a parallel phenomenon, albeit in reverse. When prompted in Japanese about pumpkin colors, GPT-4o responds with "orange", revealing what we might call an English pattern of use. When explicitly asked about pumpkin colors in Japan, however, it accurately answers "green" (see Appendix Fig. 5 and Fig. 6). This disparity suggests that cultural knowledge, while present in these models, may not naturally surface during multilingual interactions. We dub this phenomenon the *explicit–implicit localization gap*.

The distinction between explicit and implicit prompts reflects two ways models are used. In the explicit setting, we directly measure the capabilities of the model, and in the implicit setting, we measure how real users of, e.g., online chatbots, are likely to prompt the LLM. When an LLM is incapable of adapting its generation depending on the language of the input prompt, then it risks rendering many of its generations "inappropriate" for cultural context (Leibo et al., 2024). Accordingly, understanding how LLMs culturally localize is important, because it will reveal the mechanisms behind how models adapt to different cultures, which can then be used to improve its appropriateness. Quantifying the gap between implicit and explicit localization, on the other hand, contextualizes many existing cultural benchmarks — a lot of which focus on directly measurement capabilities through explicit prompts and thus may not transfer to how people actually use them.

**Contributions.** In this paper, we start by designing a simple cultural localization benchmark and quantify the explicit–implicit localization gap. We find that *implicit prompting*, using the language alone, is insufficient for an LLM to culturally localize its generation and results in a significant degradation in performance across cultural tasks. *Explicit*

*prompting*, also called "cultural prompting" (Tao et al., 2024), results in a significant jump in performance on cultural tasks but comes at the cost of increased homogeneity and stereotypicality in open-ended generation.

Next, using tools from mechanistic interpretability, we explore where cultural localization happens within LLMs and propose a method to overcome the weaknesses of explicit prompting by steering generations to be culturally localized. To this end, we identify the most relevant layers for localization using an activation patching based analysis (Vig et al., 2020; Ghandeharioun et al., 2024; Dumas et al., 2025) and compute linear steering vectors (Panickssery et al., 2024) on these layers that allow us to culturally localize model responses without the need for explicit prompting. Our experiments demonstrate that these cultural steering vectors are effective at culturally localizing generations, as well as being both *language* and *task* agnostic: they remain effective across different languages and can generalize from one localization task (such as identifying a person's name) to another (such as determining their city of origin). Steered responses also retain the diversity of the implicit prompting setting, reduce stereotypes (oversimplified, fixed beliefs or clichés about a culture) and are more faithful (accurately representative) to the underlying culture. We also find preliminary evidence for a universal cultural localization vector that steers model outputs toward the culture associated with the language of the prompt. This implies that mechanisms underlying cultural localization are also *culture-agnostic*.

As language models become increasingly widely used, understanding their true multilingual capabilities and limitations is crucial. Our work introduces the explicit–implicit localization gap as a metric for evaluating these systems, providing both a conceptual framework and practical tools for measuring cultural competence in real-world deployments. By revealing the underlying mechanisms of cultural localization, we not only advance our understanding of these models but also provide concrete paths toward building more culturally aware AI systems that better serve diverse global users. Our findings reveal that cultural localization is not as simple as explicitly telling a language model the context of the user; instead, we need to think about new approaches for culturally localizing answers.

## 2 MATERIALS AND METHODS

First, we define what we mean by language, culture, and task. By language, we refer to the language of the input prompt fed into a language model. Culture represents the beliefs, values, traditions, practices associated with a specific group of people, following the symbolic, discursive approach that has become dominant in modern anthropology (Geertz, 2017). Finally, tasks are various sets of culturally relevant problems.

### 2.1 CULTURAL LOCALIZATION BENCHMARK

We limit our analysis to English, Turkish, Russian, French, and Bengali — to cover both typographically diverse alongside high- and low-resource languages. To evaluate cultural localization in multilingual language models, we introduce a new benchmark consisting of four datasets: Names, Cities, o1-distilled, and CulturalBench (Chiu et al., 2024). The first three datasets are synthetically generated, while CulturalBench is sourced from prior work. The names dataset consists of paired names, one American and one from the country of the language; similarly, the cities dataset consists of two cities, one American, one from the country of the language we're evaluating. The o1-distilled dataset comes from few-shot prompting o1-preview (OpenAI, 2024c) to generate culturally relevant questions. We summarize the dataset in Appendix A.2.

We define cultural localization as the process of tailoring responses so they align with the cultural norms, values, and context implied by the language of the input. For example, if a cultural question is made in Bengali, a culturally localized would answer it in a way that remains faithful to Bengali culture, rather than defaulting to a response rooted in another dominant culture, like American. In the case of asking "what's a likely name here", the model should correctly identify that the name "Mohammed" is more likely than "George". It is worth noting that while language is an important indicator of cultural context, it is an often noisy proxy for a wider phenomena. For this analysis, we will treat language and culture as closely associated by assuming that language embeds within it a world of culturally-specific associations, following recent demonstrative work in large-scale corpus linguistics (Lewis et al., 2023).

### 2.2 EVALUATION

We evaluate cultural localization across the four settings illustrated. For each task, we vary the prompting language between English and Turkish/Russian/French/Bengali as well as whether we include explicit cultural context. Within this setup, we term variants including explicit cultural context in the prompt as *explicitly localized* and model performance on such prompts as their *explicit localization performance*. Variants in which cultural information is only encoded via

the language of the prompt we term as *implicitly localized*, and their corresponding performance as *implicit localization performance*.

We thus conduct two transformations on each of our data rows. First, we prepend an explicit localization text of the form "I live in [X]", where [X] is a country that speaks the language. Second, we translate both versions (with and without explicit localization) into the language of study (see Appendix A.2 for further dataset details).

**Models.** We evaluate seven models: Aya-8b-expanse (**?**) (denoted Aya-8b-it), Gemma2-9b-it, Gemma2-27b-it (Team, 2024), Llama-3.1-70b-it, Llama-3.1-8b-it, Llama-3.1-8b-base (Meta AI, 2025), and GPT-4o (OpenAI, 2024b). With the exception of open-ended generation, we sample at a temperature of 0. For open-ended generation, we use a temperature of 1, top-$p$ of 0.9, and top-$k$ of 50. For the mechanistic interpretability analysis, we study Gemma2-9b-it using nnsight (Fiotto-Kaufman et al., 2024).

**Accounting for order bias.** The order in which options are presented to a language model has been shown to affect a language model's ability to answer questions (Davidson et al., 2024; Panickssery et al., 2024). Because our benchmark consists of two options for each question, we do two passes over the dataset and average results over possible orderings.

## 2.3 ANALYZING HOW LOCALIZATION OCCURS

**Background on language modeling.** When a sentence $s$ is fed into a language model, the text is first tokenized into a sequence of tokens, $x = x_1, ..., x_t \in V$, where $V$ is the vocabulary of the model. Then when generating the $x_{t+1}$ token, the model, $P : \mathcal{X} \to \mathcal{Y}$ maps the input sequence to a probability distribution $\mathcal{Y} \in \mathbb{R}^{|V|-1}$. This is done by first embedding each of the input tokens $x_i$ using a learned input embedding matrix $E$ that maps the vocabulary to a set of embeddings denoted by $h_i^{(0)} = Ex_i$. As the token is processed throughout the model, each token's latent representation is updated as follows:

$$h_i^{(j)} = h_i^{(j-1)} + g^{(j)}(h_1^{(j-1)}, ..., h_i^{(j-1)})$$

Here $g^{(j)}$ is a function that typically consists of a causal attention layer followed by an MLP layer, alongside normalization layers. Finally, an unembedding matrix $W$ followed by the softmax operation is applied to convert the latent representation of the last token $h_t^{(L)}$ into a next-token probability distribution

$$P(x) = \frac{e^{Wh_t^{(L)}}}{\sum_{v \in V} e^{(Wh_t^{(L)})_v}} \in \mathbb{R}^{|V|-1}, \tag{1}$$

in which we omitted the dependence of $h_t^{(L)}$ on $x$.

**Activation Patching.** We use activation patching (Vig et al., 2020; Ghandeharioun et al., 2024; Dumas et al., 2025) to determine which layers of the LLM are the most important ones for our cultural localization tasks. Due to the causal masking applied in the attention layers, the latent representation of the $i$th input token after the $j$th transformer block always depends on all preceding tokens $h_i^{(j)} = h_i^{(j)}(x_1, \ldots, x_i)$. For notational convenience, we either omit this dependence when it is obvious from context (see above) or use the following short-hand notation $h_i^{(j)}(x)$.

Now, given a latent representation $h_i^{(j)}(x_{\text{source}})$ from a forward pass[1] on a source prompt $x_{\text{source}}$, we can patch this latent into another forward pass $h_i^{(j)}(x_{\text{target}})$ on a target prompt, effectively overwriting the embedding at that point, while observing how this changes the prediction $P(x_{\text{target}})$. We use $\tilde{P}(x_{\text{target}})$ to denote the target forward pass perturbed via activation patching.

More specifically, for our analysis we calculate the latent representations after each layer on the last token position $t_{\text{source}}$ during the source forward pass, i.e., $h_{t_{\text{source}}}^{(j)}(x_{\text{source}})$. Next, we select a target prompt $x_{\text{target}}$ and, during its forward pass, replace its corresponding latent for the last token with those from the source prompt $h_{t_{\text{target}}}^{(j)}(x_{\text{target}}) \leftarrow h_{t_{\text{source}}}^{(j)}(x_{\text{source}})$. To understand the mechanism behind implicit and explicit localization, we use source and target pair configurations differing in the culture of the answer, which we detail in Section 3.2.

**Activation Steering.** After finding where within a model cultural localization happens, we test if we can encourage localization through contrastive activation addition (CAA) (Panickssery et al., 2024). Here, a *steering vector* is

---

[1]Evaluating (1) is called forward pass.

formed by subtracting the mean latent representation of negative prompts from that of positive prompts for the attribute of interest. To build the positive and negative sets, we take the simplest possible contrast. In the explicit setting, a positive example is a hinted prompt where the model picks the culturally localized answer; the negative is the same prompt without the hint, where the model instead defaults to the non-localized choice. In the implicit setting, the positive is a translated prompt (into the target language) that elicits a localized answer; the negative is the English version of that prompt, which yields a non-localized one. We spell out these constructions in the main text. Formally, at layer $j$, the steering vector shifts the model away from behaviors in $D^-$ and toward those in $D^+$, and is defined as:

$$v^{(j)} = \frac{1}{|D^+|} \sum_{x \in D^+} h_t^{(j)}(x) - \frac{1}{|D^-|} \sum_{x \in D^-} h_t^{(j)}(x)$$

where we use index $t$ as a short hand for the last token position. This steering vector is then added to the generated token representations during inference

$$\tilde{h}_t^{(j)}(x) = h_t^{(j)}(x) + \alpha v^{(j)}.$$

In our case, the target behavior involves aligning the model's responses with a specific cultural context. Note that the original paper uses a different configuration: prompts for each pair remain identical until the final completion, at which point a token is appended as a hint to desired behavior. By contrast, we pair different versions of the same question (e.g., translated vs. English, with vs. without cultural context) and omit the final answer. This approach proved more effective and consistent for steering the model toward desired cultural context in our experiments.

### 2.4 EXPLICIT–IMPLICIT LOCALIZATION GAP

Using language (English vs. other) and context (with explicit prompting vs. without), we define our first explicit–implicit gap as:

$$\text{EI-Gap} = \mathbb{E}_{(x_q, y)}[\mathbf{1}\{\hat{y}(x_{\text{context}} \circ x_q) = y\}] - \mathbb{E}_{(x_{\text{tr}.q}, y_{\text{tr}})}[\mathbf{1}\{\hat{y}(x_{\text{tr}.q}) = y_{\text{tr}}\}].$$

in which $\mathbb{E}$ denotes an expectation over dataset items, $\mathbf{1}\cdot$ is an indicator for whether the model's prediction exactly matches the correct culturally localized answer, $\hat{y}(\cdot)$ is the model's produced output, $x_{\text{context}} \circ x_q$ is the explicitly localized prompt, and $x_{\text{tr}.q}$ is the implicitly localized translated prompt. The first term reflects average correctness under explicit localization, while the second term reflects average correctness under implicit localization.

**Open-Ended Generation.** To test for the potential downstream effects of explicit localization, we focus on the diversity, stereotypicality, and faithfulness of generations produced. Specifically, we create 24 short story prompts (see Appendix B) and then resample these 30 times for each story. We then calculate the cosine similarity between the embeddings of the generations in a BERTScore-style approach (Zhang et al., 2020). Specifically, we embed each generation using OpenAI's `text-embeddings-3-small` model and measure the cosine similarity across generations from each translated prompt with and without the explicit localization prompt. If the average similarity is higher across answers, we consider those answers to be more semantically similar. We also use LLM-as-a-judge in an arena-style evaluation to rank which generation is more stereotypical and faithful to the underlying culture (see Appendix B.3).

## 3 RESULTS

### 3.1 EXPLICIT–IMPLICIT LOCALIZATION GAP

We begin by quantifying the explicit–implicit localization gap within LLMs by comparing the difference in their performance across implicit and explicit settings. We illustrate the results in Figure 1 across language and model. Each cell value consists of the difference between explicit localization (English with cultural context) and implicit localization (local language with no context). We see that across most models, the gap in performance is usually above 10% and sometimes as high as 68% in the case of Bengali and Aya-8b (an explicitly multilingual model). Additionally, it appears that smaller models suffer more from the gap than larger models. The gap in Gemma2-9b-is 35% whereas in Gemma2-27b-it it is 22%. Similarly, in Llama-8b-it it is 37% whereas in Llama-70b-it it's 20% (see Appendix C for a subtask breakdown). Another curious result is that all models seem to perform poorly when prompted just in English, as evidenced by the "United States" row — English with USA context vs. just English. We hypothesize this is a consequence of the universality of English. In this way, explicitly localizing a prompt to the USA might narrow the output distribution and lead to different responses. This effect merits further investigation.

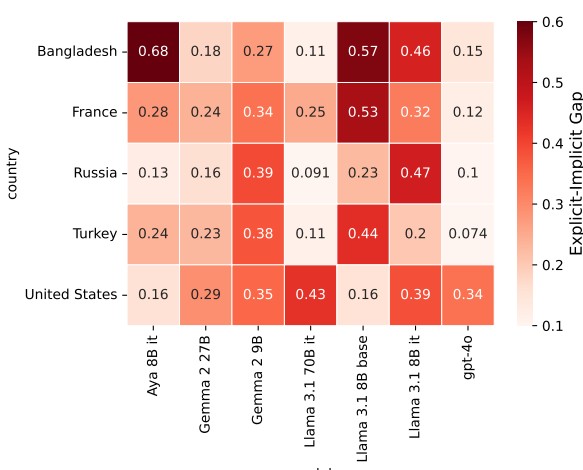

Figure 1: Heatmap showing the explicit–implicit localization gap across models and languages.

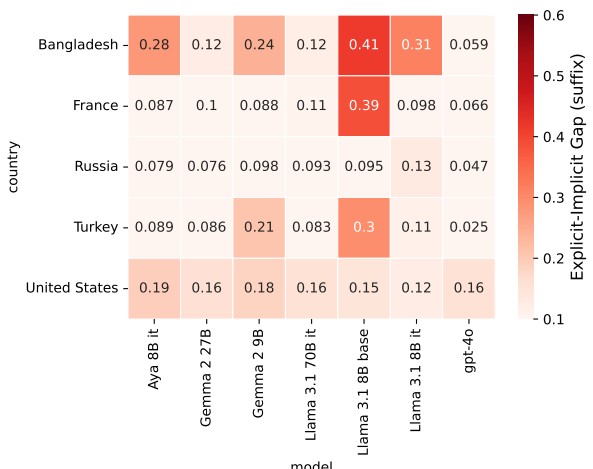

Figure 2: Heatmap showing the explicit–implicit localization gap across models and languages with a culturally relevant prefix prepended.

**Qualitative exploration of failure modes.** Our analysis includes cases where models failed to produce valid answers. While most models had failure rates below 1%, GPT-4o exhibited a notably higher rate. This was primarily due to its tendency to refuse answering questions without cultural context, particularly in name and city-related tasks. Specifically, GPT-4o declined to answer 64% of city-related and 32% of name-related questions without explicit cultural context. This cautious behavior suggests the model's built-in safeguards against potential cultural biases, though these refusals decreased somewhat when prompting in the target language.

**Characterizing explicit localization.** One natural question that arises is what text is sufficient to explicitly localize the model. Including the background of the user or language they speak seems natural. But what about more subtle approaches like including a concept unique to that culture, e.g., "omakase" for a Japanese speaker. In this section we prepend words from specific cultures and see if these words lead to comparable gains to explicit localization. We test this by including four possible words for each culture: the name of a local dish, currency, city, and cultural object (see Appendix C). In Figure 2, we show the same difference between explicit and implicit localization as above, only now with the implicit being the correctness with a random prepended cultural word. We observe that across all models and languages, the gap falls dramatically. This implies that having a culturally specific word in the prompt is enough to localize the model to that culture.

**Effect on Open Generation.** A problem with prompting approaches is that they can reduce entropy in the corresponding generation by giving the model context on what to generate (Chu et al., 2025) and unfaithfully simulating ethnic groups (Wang et al., 2025). In this question, we explore potential downsides with explicit localization: homogeneity and stereotypicality. Specifically, we take open-ended generation prompts that are each translated into the languages, and generate thirty generations for each using a temperature of 1 across explicit and implicit settings. Note that we drop the base models for this task.

In Table 1, we show the homogeneity in the implicit and explicit settings. We find that across all models, homogeneity goes up (except for Aya, which was not evaluated). This is most notable in the case of the Gemma-2 models, where it increases on average by more than 3%. Additionally, in Figure 14 we show that explicit prompting results in more stereotypical answers than implicit prompting.

## 3.2 PINPOINTING CULTURAL LOCALIZATION

We now examine a deeper explicit–implicit gap based on how language models culturally localize text in the implicit and explicit setting through activation patching. For the implicit localization analysis, we construct pairs of prompts: a target prompt in English that generates a non-localized response, and a source prompt that is its translation with modified option labels (using letters A, B instead of numbers). This modification of option keys helps isolate the source prompt's influence on model behavior. Concretely, the English version may look like "A common name here is (1)

|  | Implicit | Explicit |
|---|---|---|
| Gemma 2 27B | $0.333 \pm 0.008$ | $0.359 \pm 0.006$ |
| Gemma 2 9B | $0.337 \pm 0.007$ | $0.368 \pm 0.006$ |
| Llama 3.1 70B it | $0.298 \pm 0.006$ | $0.323 \pm 0.006$ |
| Llama 3.1 8B it | $0.301 \pm 0.006$ | $0.332 \pm 0.006$ |
| GPT-4o | $0.317 \pm 0.006$ | $0.324 \pm 0.006$ |

Table 1: Global cosine similarity across generations. Bootstrap standard errors on the means are shown.

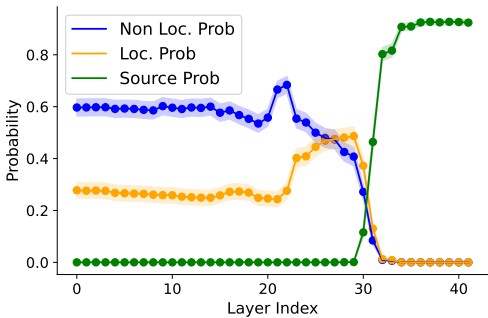 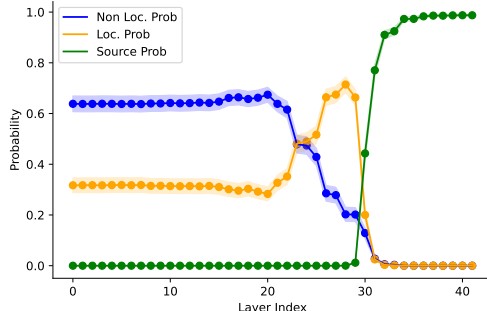

Figure 3: Activation patching results, where target prompt localized token probability (Loc. Prob) is shown in yellow, and non-localized target prompt token probability (Non Loc. Prob) is shown in blue. Finally, green shows the probability of answering the question from the source prompt. Shaded regions around plot lines represent 95% confidence intervals (CI), calculated as mean $\pm 1.96 \times$ SEM. (Left) Source-translated prompt and target English prompt. (Right) Source-translated prompt with cultural context and target non-context translated prompt.

George, (2) Sergey. Answer:" where the model would answer (1) for George. The Russian copy would be a mere translation of this.

For explicit localization, we use only translated examples, because they show the most consistent localization when given cultural context. Here, we pair target prompts without context with their counterparts including explicit cultural context as source prompts, again using modified option labels. In this case, the text we pass in would not be the translated version of the prompt above, but instead one with "I live in the United States." as a prefix.

Figure 3 presents three probability trajectories across model layers: the likelihood of decoding the non-localized response from the target prompt (Non. Loc. Prob), the probability of culturally localizing the target prompt to the locality of the source prompt (Loc. Prob), and the probability of generating the modified option character (A, B instead of 1, 2) from the source prompt (green line). This modification helps us identify the layers at which the source prompt starts to dominate the model's behavior.

Substituting the layer early leads the context from the target prompt to effectively "overwrite" latent information from the passed-in source. When we pass it across later layers, the generation is almost always simply the source probability because the attention unable to write the relevant context. The interesting points are in the middle, where we see the correct localized answer (from the target forward pass) spike. This means that the source latent is writing in the cultural localization answer *without* overwriting the answer.

These results indicate: (1) Implicit and explicit localization spike and drop at the same layer, namely 23 and 30, suggesting that the mechanisms behind cultural localization may indicate the consolidation of a world model that becomes culturally customized within these specific middle layers. We decompose our analysis across languages and tasks, and find a similar pattern (see Appendix F). (2) Implicit localization is less pronounced than explicit localization, implying that language may not contain sufficient context for the language model to culturally localize.

### 3.3 STEERING LOCALIZATION

Following from our activation patching approach, we design a steering experiment by extracting embeddings from two types of prompts: ones with explicit localization (e.g., "I live in Turkey") and control prompts without cultural context. To extract steering vectors, we use a different prompt structure, detailed in Appendix D. We subtract the control embeddings from their localized counterparts, performing this process in both English and translated versions. When

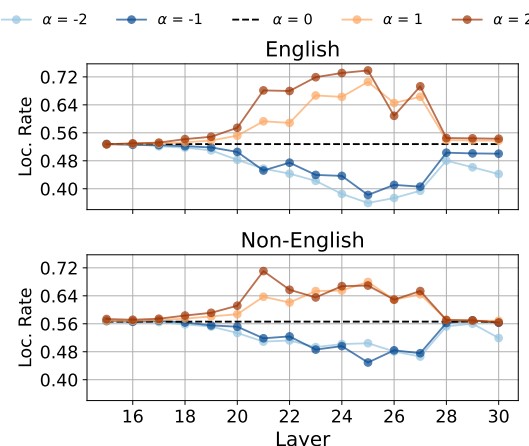

Figure 4: Steering results for per-culture vectors calculated using English pairs with $\alpha \in [-2, 2]$ across layers [15-30], where the horizontal axis represents the layer at which the steering vector is applied, and the vertical axis indicates the ratio of localized responses. Titles denote prompt language.

|  | $\boldsymbol{v}_{\text{en}}$ | $\boldsymbol{v}_{\text{tr.}}$ | $\boldsymbol{v}_{\text{names}}$ | Implicit | Explicit |
|---|---|---|---|---|---|
| cities | $0.873 \pm 0.021$ | $0.740 \pm 0.027$ | $0.896 \pm 0.019$ | $0.466 \pm 0.032$ | $1.000 \pm 0.000$ |
| culturebench | $0.793 \pm 0.042$ | $0.780 \pm 0.042$ | $0.798 \pm 0.042$ | $0.742 \pm 0.044$ | $0.882 \pm 0.034$ |
| culturedistil | $0.614 \pm 0.029$ | $0.575 \pm 0.031$ | $0.624 \pm 0.029$ | $0.484 \pm 0.031$ | $0.844 \pm 0.023$ |
| names | $0.734 \pm 0.026$ | $0.680 \pm 0.028$ | $0.771 \pm 0.024$ | $0.552 \pm 0.028$ | $0.914 \pm 0.016$ |

Table 2: Steering performance across subtasks using $v_{\text{names}}$, $v_{\text{en}}$, and $v_{\text{tr.}}$; details on the alpha and layers in Appendix D. Each value is the fraction of answers that are culturally localized. Implicit shows the cultural localization rate in the implicit context. Explicit shows the rate when context is directly provided.

we use all tasks (names, cities, o1-distilled, CulturalBench) we denote these two settings as $v_{\text{en}}$ and $v_{\text{tr.}}$, respectively. Alternatively, when we steer only using the translated names dataset, we use $v_{\text{names}}$. Using these steering vectors, we run three analyses: (1) How much of the underlying explicit prompting performance can be recovered by adding a single vector? (2) To what extent is the steering vector task agnostic? (3) Are steering vectors culture-specific, or can a steering vector from one culture be used to steer another culture?

**Steering performance.** In Figure 4 we show the steering performance across layers and alpha values. The top plot includes results for the English language setting, and in the bottom plot, we show what happens when we apply the English steering vector on other languages. We observe that in both settings, steering around layer 19 begins to increase localization and continues until layer 28, following a similar pattern to the activation patching above. Interestingly, we find that the English steering vector generalizes to other languages. For example, adding the English steering vector to a different language leads to an impressively improved rate of localization. Subtracting the vector leads to a drop in localization with answers more aligned with the United States. We also provide a breakdown of the performance across tasks in Table 2.

In this section, we only show results for steering on the explicit prompts. An alternative way to design steering experiments is through translated prompts and their English variants. For example, take the Russian prompt (no explicit localization) and subtract its English counterpart. This approach, however, is less effective (see Appendix D).

**Steering across tasks.** Given a cultural localization vector from one task, we next show it can be generalized to other tasks. For example, if we calculate the localization for Russian using the names subtask, it can be used to localize in the CulturalBench dataset. To study this question, we extract task-specific cultural localization vectors and apply them to different tasks. In Table 2, we show that steering using the names dataset gives comparable results to steering on a specific task. In general, the difference in performance is low, with the exception of cities where we find that the name task leads to a better performance than task-specific steering. This implies that cultural localization is task agnostic and represents a much broader, more abstract phenomenon present in large language models.

|  | Implicit | Explicit | $v_{\text{tr},\alpha=2}$ |
|---|---|---|---|
| Bengali | $0.367 \pm 0.006$ | $0.400 \pm 0.005$ | $0.359 \pm 0.005$ |
| English | $0.285 \pm 0.009$ | $0.339 \pm 0.006$ | $0.286 \pm 0.007$ |
| French | $0.341 \pm 0.007$ | $0.374 \pm 0.006$ | $0.352 \pm 0.006$ |
| Russia | $0.265 \pm 0.007$ | $0.312 \pm 0.006$ | $0.283 \pm 0.008$ |
| Turkey | $0.396 \pm 0.007$ | $0.434 \pm 0.005$ | $0.411 \pm 0.007$ |

Table 3: Homogeneity results for Gemma2 9b it in the implicit, explicit, and steered ($v_{\text{tr}}$, $\alpha = 2$, layer 25). Bootstrap standard errors are shown on the means.

|  | $v_{\text{en}}$ | $v_{\text{tr}}$ | $v_{\text{names}}$ | $v_{\text{universal (tr.)}}$ | Implicit | Explicit |
|---|---|---|---|---|---|---|
| Bangladesh | $0.739 \pm 0.036$ | $0.789 \pm 0.035$ | $0.742 \pm 0.036$ | $0.692 \pm 0.040$ | $0.628 \pm 0.040$ | $0.894 \pm 0.026$ |
| France | $0.612 \pm 0.041$ | $0.648 \pm 0.040$ | $0.662 \pm 0.038$ | $0.586 \pm 0.040$ | $0.547 \pm 0.045$ | $0.884 \pm 0.028$ |
| Russia | $0.793 \pm 0.032$ | $0.755 \pm 0.036$ | $0.802 \pm 0.034$ | $0.690 \pm 0.040$ | $0.532 \pm 0.041$ | $0.877 \pm 0.029$ |
| Turkey | $0.763 \pm 0.030$ | $0.704 \pm 0.033$ | $0.783 \pm 0.030$ | $0.680 \pm 0.036$ | $0.559 \pm 0.037$ | $0.919 \pm 0.020$ |
| United States | $0.695 \pm 0.028$ | $0.695 \pm 0.028$ | $0.720 \pm 0.026$ | $0.610 \pm 0.028$ | $0.524 \pm 0.029$ | $0.883 \pm 0.019$ |

Table 4: Results for various steering vectors across cultures. Values show the fraction of questions that are correctly culturally localized for each culture (row). $v_{\text{universal (tr.)}}$ refers to held-out universal steering vector, where we average over all culture-specific translated steering vectors with the exception of the culture.

**Open-ended generation.** In the previous section, we showed that model steering can be an effective approach for culturally localizing a language model. We now test the effect of steering on open-ended generation. In Table 3 we show the homogeneity of generations for Gemma-2-9b-it in implicit, explicit, and steered settings ($v_{\text{tr}}$) across languages. We observe that steering results in more diverse outputs across generations than adding explicit context, but only slightly less diverse than implicit prompting. In Appendix B we show examples of our generations and note that steering causes the model to generate in a format similar to the implicit prompt, but with culturally-relevant information. We also evaluate stereotypicality and faithfulness and find that the steered model is both less stereotypical than the explicit setting, and more faithful to the appropriate cultural context (see Appendix B.3).

**Universal culture vector.** In our final analysis, we test where cultural localization is universal: does there exist a vector one can add to prompts in any language that automatically localizes it to the language of that prompt? To study this question, we calculate the "universal" steering vector by taking the average of all culture-specific steering vectors, except for the language we are currently evaluating, and then apply the vector to the held-out language and measure its effect on cultural localization. We do this only in the translated context where we take explicitly localized prompts in all the languages (except for the one we're currently studying) and subtract the implicit prompt.

In Table 4, we observe that universal steering does lead to an improvement over implicit prompting alone, but still falls behind explicit prompting. It is also worse than steering on the culture of the prompt alone, as evidenced by the $v_{\text{tr}}$ vector. Despite this, the fact that universal steering directionally leads to improved cultural localization suggests the existence of a way to universally steer the model to generate culturally relevant answers.

In the current construction of the task, we limit our analysis to binary questions where one of the answers is always from the United States. In this setting, the cultural steering vector may simply learn to provide the "non-American" answer. To account for this, we extend to a multiple choice setting where each of the options comes from a different culture. We notice a similar improvement from universal steering in this context (see Appendix E). We leave it to future research to determine how best to define a universal steering vector or enable model soft control over language-specific vectors, but here provide an initial support for the existence of automatic cultural customization mechanisms.

## 4 RELATED WORK

**Multilingual evaluations in LLMs.** Many early multilingual benchmarks like MMLU (Dac Lai et al., 2023), math (Shi et al., 2022), commonsense tasks (Lin et al., 2021; Ponti et al., 2020), factual knowledge (Kassner et al., 2021; Zhou et al., 2022), and representations (Conneau et al., 2018), are based on translations of English benchmarks and thus by construction measure mostly culture-agnostic capabilities. Recently, there have been several attempts to address this shortcoming by creating benchmarks aimed at explicitly measuring cultural capabilities (Chiu et al., 2024; Yin et al., 2022; Naous et al., 2024; Singh et al., 2025). The evaluation methodology in these works corresponds to our

explicit setting. Implicit prompting has also been studied (Vayani et al., 2024; Zhang et al., 2023; Arora et al., 2023). One notable work is by Romanou et al. (2024), where they collect real exam questions from 44 written languages and evaluate capabilities in a local context using these questions. Other work has extended multilingual analysis into reward modeling (Gureja et al., 2024) and multimodal models (Kannen et al., 2025; Khanuja et al., 2025). Tao et al. (2024) showed that LLMs are culturally biased in values and propose using "cultural prompting" to reduce that bias; similarily (AlKhamissi et al., 2024) argued in favor of "anthropological prompting". There is also active research focusing on knowledge transfer across languages (Goldman et al., 2025; Rajaee & Monz, 2024) and answer consistency (Qi et al., 2023). As far as we know, our work is the first that attempts to rigorously measure the difference in performance between the explicit and implicit setting, and propose techniques outside of prompting to limit the gap.

**Multilingual language models.** The growth in LM applications has led to a corresponding increase in research into designing multilingual models (Conneau & Lample, 2019; He et al., 2024; Muennighoff et al., 2023; Üstün et al., 2024). Work has created instruction tuning datasets (Singh et al., 2024) to align multilingual models and studied the impacts of multilinguality on overall model performance (Chang et al., 2023; Schäfer et al., 2024; Gurgurov et al., 2024; Held & Yang, 2022). There has been a large country-level push to train multilingual models (Piir, 2023; Martins et al., 2024; OpenAI, 2024a; Hornyak, 2023; Yue et al., 2025).

**Mechanistic interpretability for multilinguality.** Techniques like activation patching (Meng et al., 2023; Ghandehari-oun et al., 2024; Vig et al., 2020), sparse autoencoders (Huben et al., 2024), and steering (Panickssery et al., 2024) have been used to study linguistic representations. These techniques have shown that linguistic bias seems to originate from a deeper representational problem with the underlying concepts biased towards English (Wendler et al., 2024; Alabi et al., 2024; Wu et al., 2024; Schut et al., 2025). Further, it has been shown that in addition to having concept representations biased towards a specific culture, the circuits[2] the model uses to complete multilingual tasks are often language-agnostic (Lindsey et al., 2025; Wang et al., 2024; Held & Yang, 2022). Other work has argued that instead of concepts being biased towards English, they really occupy a universal language-agnostic representation (Brinkmann et al., 2025; Dumas et al., 2025; Variengien & Winsor, 2023). Finally, prior work found that language models have specific neurons responsible for generating in a specific language (Tang et al., 2024).

## 5 DISCUSSION

In the introduction, we motivated cultural localization through the example of pumpkin color, but this phenomenon stretches far beyond simple concept features. Culture forms the way people think about the world from freedom to religion, from happiness to money (Haerpfer et al., 2020; Hofstede, 1984). Many of these cultural associations lie embedded within language (Lewis et al., 2023; Lakoff, 2008). While our benchmark focused on simple examples (like names or cultural facts) it demonstrates strong evidence that language alone is insufficient to culturally localize language models. As LLMs become used around the world and in many different cultural contexts, the lack of adaptation can lead to unfaithful answers inappropriate for context. Addressing this limitation is key to creating truly global models.

Similar to past work (Tao et al., 2024), we find that providing explicit context helps confront some of these issues. We build on that and show that explicit prompting is not a panacea, however, but itself comes at the cost of increased homogeneity and stereotypicality. Nevertheless, we discover that large models contain the capability for cultural specificity within their multi-layered representations and we locate to mechanically unlock that capacity. We show that an approach focusing on model internals can reconcile this problem, demonstrating how even simple model steering via the addition of a single vector at a single layer is able to culturally localize model answers. While steering in this way did not recover the full cultural localization performance of adding explicit context to the prompt, its cultural localization is more culturally faithful, less stereotypical, and more diverse (see Appendix B). We think that this distinct behavior is a result of the steering intervention being more surgical, mechanistically explicit, and having a more localized effect on the forward pass. We add the steering vector only for newly generated tokens and only at a relatively late layer, leaving the remaining forward pass unchanged. In contrast, explicit prompting modifies the forward pass in its entirety.

This distinction between explicit approaches (providing context directly) and implicit approaches (relying on language alone) highlights a broader conceptual divide in how we evaluate cultural localization. We believe that both explicit and implicit localization are important constructs to measure. But it is critical to know what they involve and represent. Whereas the former predominantly measures capabilities, the latter is ecologically valid (Brunswik, 1940). This difference between evaluations for model developers and those informative for downstream users is often underappreciated (Kapoor et al., 2024), and as we argue in this paper, it is also true for multilingual benchmarks. Explicit localization

---

[2]A circuit is a subnetwork within a neural network explaining most of its performance on a specific task. Tasks can be defined, e.g., via an input-output dataset.

measures whether we can steer models towards outputting faithful content, but it tells us little about how models behave in real-world settings. Others study this problem purely in an implicit setting, and thus come to conclusions like "Don't trust ChatGPT when your question is not in English" (Zhang et al., 2023) due to biased representations. While this does indeed appear to be true (Wendler et al., 2024; Dumas et al., 2025), it erroneously implies that this knowledge does not exist within language models. We recommend that multilingual benchmark developers become aware of these two settings and clearer about what they study. When possible, they should develop evaluations for both.

## LIMITATIONS

There are several limitations to our study. First, we do not measure the downstream effects of prompting and steering outside of simple model-based approaches (embedding similarity and LLM-as-a-judge). While we include a few anecdotal examples in the appendix, a more thorough exploration should be undertaken, ideally run through a user study with people from these cultures. Second, we do not focus on exploring the many possible strategies for guiding a language model's generations to relate to a specific culture. Future work should explore: (a) different methods for computing steering vectors like distributed alignment search (Geiger et al., 2024; Minder et al., 2024), affine steering (Marshall et al., 2025), or dictionary learning (Huben et al., 2024) and (b) methods for adapting the model through parameter efficient finetuning methods (Li & Liang, 2021; Dettmers et al., 2023; Hu et al., 2021; Wu et al., 2025; Houlsby et al., 2019). Third, we do not rigorously evaluate the universal steering vector. While we find evidence that it exists, we leave it for future work to rigorously study it and its downstream implications. Fourth, we limit our analysis to language models and do not consider the current paradigm of reasoning models. Future work should extend the results to both and explicitly trained reasoning models and multimodal models, which have been shown to exhibit poor multilingual performance (Bansal et al., 2022; Kannen et al., 2025). Next, our findings also rely on specific prompt designs, and results may vary with different formulations or personalization mechanisms (Biderman et al., 2024). We only evaluate five languages, leaving open questions about how cultural localization behaves across a broader range of linguistic and cultural settings (Khanuja et al., 2024). Finally, we note that we assume a direct mapping between culture and language. This may not be true and future work should better disentangle their connection in language models.

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

# A APPENDIX

## A.1 MOTIVATING EXAMPLE

In the introduction, we share an example of how when you ask GPT-4o what color a pumpkin is in Japanese it states orange; whereas, when explicitly asked it says green. We include those examples here.

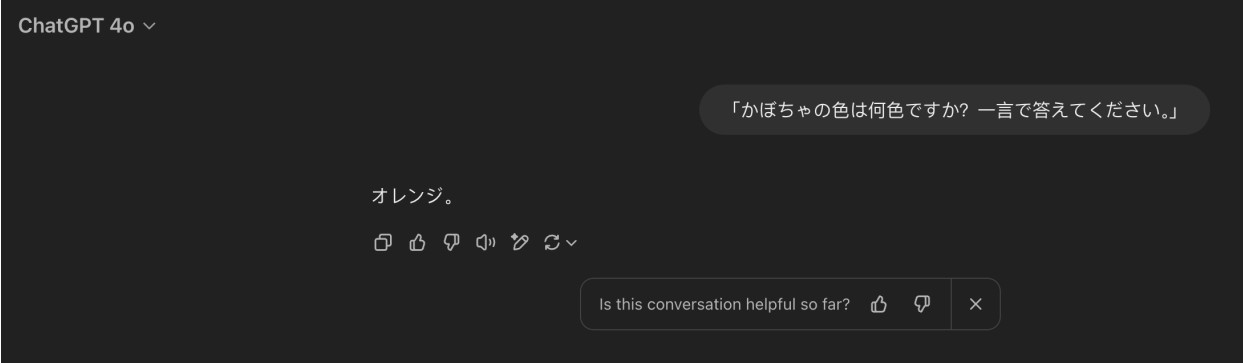

Figure 5: When prompted with "What color is a pumpkin. Please answer in one word." Model answers orange.

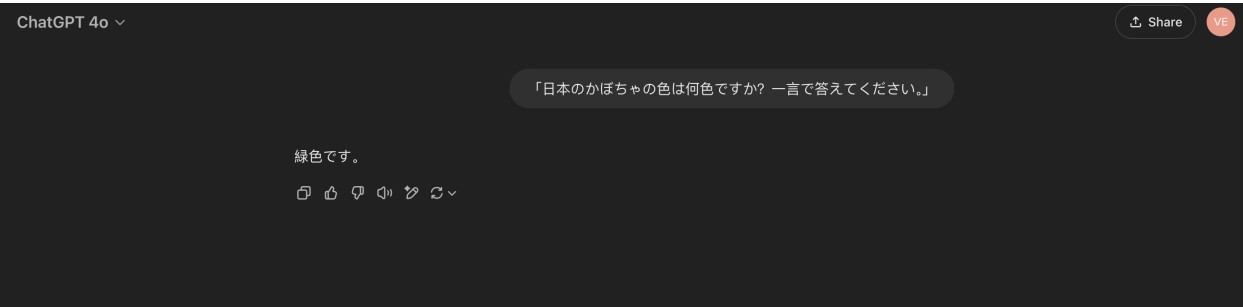

Figure 6: When prompted with "What color is a pumpkin in Japan. Please answer in one word." Model answers green.

## A.2 CULTURAL LOCALIZATION BENCHMARK

1. **Names:** We selected the 100 most common male and female names for each country and paired names of the same gender with a maximum difference of 2 in their popularity rankings. We then posed the question: "Which of the following names is most likely to be my name?" The names were sourced from the *Behind the Name* website, which reportedly bases its data on official statistics (Behind the Name, 2025). Note that the website does not contain entries from Bangladesh, so we use GPT-4o to generate these names.

2. **Cities:** We selected the 20 most populated cities (where applicable) in each country and formulated the question: "In which city am I most likely to live?" The list of cities was obtained from the respective country's Wikipedia page.

3. **o1 Distillation:** We provided a few demonstration examples related to food, drinks, traditions, and laws specific to a given country. Using these examples, we prompted the OpenAI `o1_preview` model to generate 50 additional questions of a similar nature (OpenAI, 2025).

4. **CulturalBench:** We use the dataset from (Chiu et al., 2024) with the Easy setting, where we filter out questions with multiple correct answers and binarize the remaining ones by pairing each correct answer with three incorrect options.

Table 5: Breakdown of the Cultural Localization Benchmark dataset (transposed). Rows show each subtask, while columns show the countries. The final row and column indicate totals across columns and rows, respectively.

Example prompts from datasets in Figure 7.

---

**Example prompts from different datasets:**

**Names Dataset:** Which of the following names is most likely to be my name?
Options:

1. Sergey
2. Bill

**Cities Dataset:** In which city do I most likely live?
Options:

1. Chicago
2. Istanbul

**o1-distilled Dataset:** Around here, what is the most common alcoholic drink?
Options:

1. Rakı
2. Whiskey

**CulturalBench Dataset:** Which festival celebration style is most likely appropriate for expressing the collective cultural identity?
Options:

1. Hosting a potluck where each individual brings a dish from their own ethnic background.
2. Celebrating only the most widely recognized national holiday.

---

Figure 7: Example prompts from the *Cities*, *o1-distilled*, and *CulturalBench* datasets.

## A.3 TRANSLATION AND COUNTRY REMOVAL

We translate all prompts using GPT-4o (Achiam et al., 2024) and the following prompt:

---

Translate the following text from {src_lang} to {tar_lang}. Respond with just the translation, no additional information.

TRANSLATE: {text}

---

Figure 8: Prompt used for translation.

For CulturalBench, we remove all mentions of countries using the following prompt also using GPT-4o:

---

From the following text, remove any explicit mentions of a country.

For example: "In India, what is a common meal to have for lunch?" will become "What is a common meal for lunch?"

Only respond with the rewriting.

TEXT: {text}

---

Figure 9: Prompt used for removing mentions of countries from CulturalBench.

When prompting the base model, we adopt a different few-shot prompt syle:

```
Question:
What am I?
Options:
1. Human
2. Robot
Answer: 1
Question:
Which planet is closest to the sun?
Options:
1. Mercury
2. Venus
Answer: 2
Question:
I live in Turkey. Which of the following names is most likely to be my name?
Options:
1. Liam
2. Alparslan
Answer:
```

Figure 10: Prompt used for generation from base model.

# B  OPEN-ENDED GENERATION

## B.1  PROMPTS

Here we include the prompts used in the open-ended generation task.

- Write me a recipe for a local dish.
- Write me a short story about a boy and his kite.
- Write me a 100-word bedtime story for kids.
- Tell me about what happened during the Titanic.
- Describe a day in the life of an astronaut on Mars.
- Explain photosynthesis as if I'm five years old.
- Write a letter from a pirate to his long-lost friend.
- Invent a new holiday and describe how people celebrate it.
- Tell me a joke that would make a robot laugh.
- Describe the feeling of standing at the edge of a cliff.
- Write a poem about a lonely lighthouse.
- Explain gravity without using scientific jargon.
- Create a dialogue between a cat and a dog arguing about dinner.
- Write a product review for an imaginary gadget.
- Describe a futuristic city 500 years from now.
- Tell me a legend about a magical forest.
- Explain how to build a sandcastle like a pro.
- Write a diary entry from the perspective of a dragon.
- Imagine you're a time traveler—describe your first day in the past.
- Give me instructions on how to be invisible for a day.
- Write a letter from Earth to an alien civilization.
- Describe a sunset without using the words 'red,' 'orange,' or 'yellow.'
- Tell me about a secret hidden inside an old library.
- Invent a sport that could be played on the moon.

## B.2  ADDITIONAL DETAILS ON HOMOGENEITY RESULTS

In this section we include homogeneity results by model and language. Refer to Table 6.

| Language | Llama-3.1-70B | | Llama-3.1-8B | | GPT-4o | |
|---|---|---|---|---|---|---|
| | Implicit | Explicit | Implicit | Explicit | Implicit | Explicit |
| bn | $0.303 \pm 0.004$ | $0.324 \pm 0.004$ | $0.295 \pm 0.004$ | $0.328 \pm 0.005$ | $0.426 \pm 0.005$ | $0.435 \pm 0.005$ |
| en | $0.235 \pm 0.007$ | $0.240 \pm 0.006$ | $0.243 \pm 0.008$ | $0.246 \pm 0.007$ | $0.240 \pm 0.007$ | $0.236 \pm 0.007$ |
| fr | $0.304 \pm 0.006$ | $0.326 \pm 0.006$ | $0.308 \pm 0.006$ | $0.334 \pm 0.006$ | $0.308 \pm 0.007$ | $0.312 \pm 0.007$ |
| ru | $0.231 \pm 0.006$ | $0.257 \pm 0.006$ | $0.247 \pm 0.006$ | $0.285 \pm 0.007$ | $0.235 \pm 0.007$ | $0.251 \pm 0.006$ |
| tr | $0.386 \pm 0.008$ | $0.436 \pm 0.008$ | $0.379 \pm 0.006$ | $0.439 \pm 0.006$ | $0.364 \pm 0.006$ | $0.377 \pm 0.006$ |

| Language | Gemma 2 27B | | Gemma 2 9B | |
|---|---|---|---|---|
| | Implicit | Explicit | Implicit | Explicit |
| bn | $0.348 \pm 0.009$ | $0.404 \pm 0.006$ | $0.368 \pm 0.006$ | $0.402 \pm 0.005$ |
| en | $0.286 \pm 0.009$ | $0.286 \pm 0.006$ | $0.286 \pm 0.008$ | $0.297 \pm 0.006$ |
| fr | $0.339 \pm 0.007$ | $0.360 \pm 0.005$ | $0.341 \pm 0.007$ | $0.375 \pm 0.006$ |
| ru | $0.274 \pm 0.008$ | $0.296 \pm 0.007$ | $0.266 \pm 0.007$ | $0.312 \pm 0.007$ |
| tr | $0.386 \pm 0.007$ | $0.417 \pm 0.006$ | $0.398 \pm 0.007$ | $0.435 \pm 0.005$ |

Table 6: Homogeneity scores for different models across languages.

### B.3 ADDITIONAL DETAILS ON STEREOTYPICALITY AND FAITHFULNESS RESULTS

To evaluate the stereotypicality and faithfulness of model outputs, we used LLM-as-a-judge (Zheng et al., 2023) with GPT-4o in an arena-style setting. Recall that in the open-ended generation, we had each model answer a prompt 30 times across the different languages in both the implicit and explicit setting. For the arena evaluation we looped over each story and sampled 15 generations in the explicit, implicit, and steered settings. We then took pairwise comparisons between explicit–implicit and explicit-steered and ask GPT-4o to rate which one is more stereotypical and faithful to the culture. Figure 11 shows the stereotypicality prompt, and Figure 12 shows the faithfulness prompt.

---

You will be given two text excerpts from different sources. I would like you to compare them based on how stereotypical they are of culture culture. Stereotypical means that a text relies on oversimplified, fixed beliefs or clichés about a culture. It reduces the complexity of a cultural group to a small number of traits—often leading to misrepresentation or reinforcing biases.
Text A: {text_a}
Text B: {text_b}
Which text relies more on cultural stereotypes about {CULTURE} people or culture? Answer with just 'A' if Text A is more stereotypical, 'B' if Text B is more stereotypical, or 'TIE' if they are equally stereotypical.

---

Figure 11: LLM-as-a-judge arena-style prompt for rating the stereotypicality of generations.

---

You will be given two text excerpts from different sources. I would like you to compare them based on how faithful they are to the culture culture. Faithful means that the text represents cultural practices, beliefs, or values in a nuanced, accurate, and respectful manner. It acknowledges internal diversity and context, avoids homogenizing or flattening a group's identity, and strives for factual correctness.
Text A: {text_a}
Text B: {text_b}
Which text is more faithful to the {CULTURE} culture? Answer with just 'A' if Text A is more faithful, 'B' if Text B is more faithful, or 'TIE' if they are equally faithful.

---

Figure 12: LLM-as-a-judge arena-style prompt for rating the faithfulness of generations.

---

You will be given two text excerpts. I would like you to compare them based on their fluency. Fluency means that the text is written in natural, grammatically correct language with coherent sentence structure and smooth flow. A fluent text is easy to read and understand.
Text A: {text_a}
Text B: {text_b}
Which text is more fluent? Answer with:
- 'A' (if Text A is more fluent)
- 'B' (if Text B is more fluent)
- 'TIE' (if they are equally fluent)
Do only answer with the letter, no other text.

---

Figure 13: LLM-as-a-judge arena-style prompt for rating the fluency of generations.

Table 7 shows the results for stereotypicality. Across all languages with the exception of French, the steered generation generates less steretypical results. We also run a cultural faithfulness evaluation and show the results in Table 8. Observe that across all languages with the exception of Turkish, the steering provides more faithful generations. Figure 14 shows the fraction GPT-4o says a specific setting is more stereotypical.

### B.4 SAMPLE GENERATIONS

Below we include sample generations. In Figure 17, 18 we show apply an American steering vector affects generation for a prompt about a recipe and a holiday. Afterwards, we show what happens when we apply each cultures steering vector onto the same English prompt — illustrating that steering is possible cross language. A few observations. First of

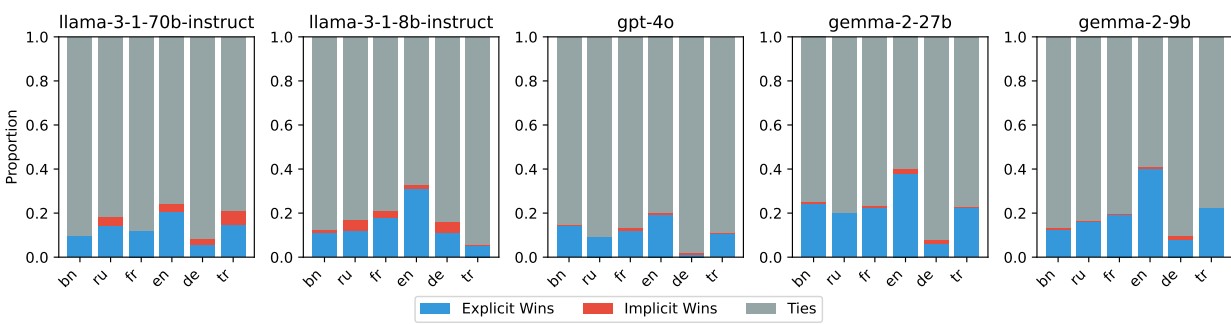

Figure 14: Stereotypicality win rate between the explicit and implicit. Winning here means that the models generation is more stereotypical.

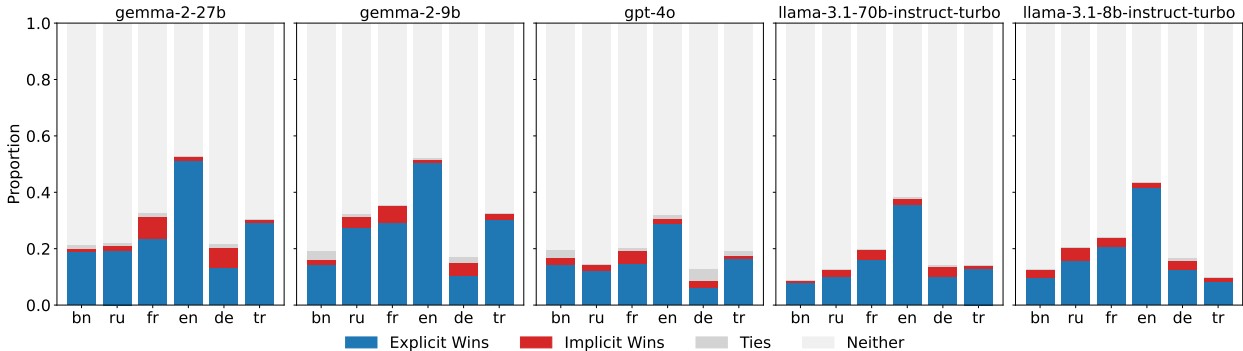

Figure 15: Stereotypicality win rate between the explicit and implicit judged by GPT-OSS-120B with reasoning effort set to medium.

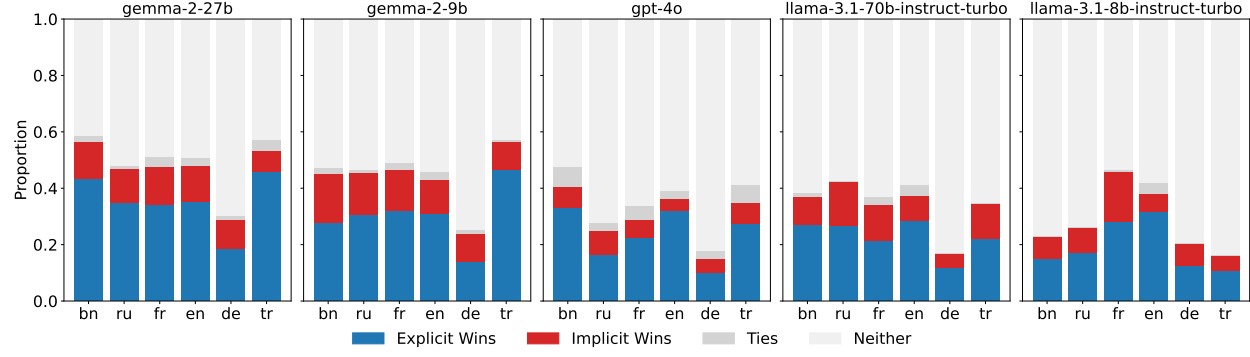

Figure 16: Faithfulness win rate between the explicit and implicit judged by GPT-OSS-120B with reasoning effort set to medium.

|  | Implicit | Explicit | $v_{\text{tr.}}$ |
|---|---|---|---|
| bn | $0.011\pm_{0.020}$ | $0.125\pm_{0.067}$ | $0.114\pm_{0.064}$ |
| ru | $0.008\pm_{0.017}$ | $0.158\pm_{0.078}$ | $0.089\pm_{0.058}$ |
| fr | $0.006\pm_{0.014}$ | $0.192\pm_{0.086}$ | $0.267\pm_{0.089}$ |
| en | $0.008\pm_{0.019}$ | $0.403\pm_{0.103}$ | $0.156\pm_{0.078}$ |
| tr | $0.003\pm_{0.008}$ | $0.222\pm_{0.083}$ | $0.103\pm_{0.058}$ |

Table 7: Stereotypicality scores for different models across languages for Gemma 2 9B. Steered using layer 25, $\alpha = 2$.

|  | Implicit | Explicit | $v_{\text{tr.}}$ |
|---|---|---|---|
| bn | $0.117\pm_{0.067}$ | $0.183\pm_{0.081}$ | $0.456\pm_{0.100}$ |
| ru | $0.083\pm_{0.056}$ | $0.206\pm_{0.081}$ | $0.250\pm_{0.092}$ |
| fr | $0.031\pm_{0.036}$ | $0.169\pm_{0.078}$ | $0.306\pm_{0.097}$ |
| en | $0.039\pm_{0.036}$ | $0.147\pm_{0.072}$ | $0.164\pm_{0.078}$ |
| tr | $0.083\pm_{0.056}$ | $0.267\pm_{0.092}$ | $0.200\pm_{0.078}$ |

Table 8: Faithfulness scores for different models across languages for Gemma 2 9B. Steered using layer 25, $\alpha = 2$.

all, the steered generations have the same syntactic structure to the implicit generation prompt, using similar generation structure. Second, the steering vector doesn't force the generation to only generate for a specific context.

| | Explicit | Steering |
|---|---|---|
| bn | $0.108\pm_{0.016}$ | $0.886\pm_{0.017}$ |
| ru | $0.269\pm_{0.023}$ | $0.667\pm_{0.026}$ |
| fr | $0.361\pm_{0.023}$ | $0.492\pm_{0.025}$ |
| en | $0.283\pm_{0.024}$ | $0.508\pm_{0.026}$ |
| tr | $0.269\pm_{0.023}$ | $0.706\pm_{0.023}$ |

Table 9: Fluency comparison. Win rates for explicit vs. steering for Gemma 2 9B across languages.

**Explicit:** Alright, let me whip up something delicious for you, something distinctly American, but with a little southern twist! We're gonna make some **Shrimp and Grits**. It's a classic Lowcountry dish hailing from the coastal areas of South Carolina and Georgia.
**Here's what you'll need:**
**For the Grits:**
* 1 cup stone-ground grits
* 4 cups water
* 1 teaspoon salt
* 4 tablespoons unsalted butter
* 1/2 cup grated sharp cheddar cheese

**For the Shrimp:**

* 1 pound large shrimp, peeled and deveined
* 1/4 cup butter
* 1/4 cup chopped onion
* 2 cloves garlic, minced

**Implicit:** ## A Taste of Tuscany: Ribollita
Ribollita, a hearty Tuscan bread soup, is perfect for a comforting meal on a chilly evening.
**Ingredients:**
* 1 tbsp olive oil
* 1 large onion, chopped
* 2 carrots, chopped
* 2 celery stalks, chopped
* 2 cloves garlic, minced
* 8 cups vegetable broth
* 1 (14.5 oz) can diced tomatoes, undrained

**Steered:** ## Texas Chili (Classic Texas Style)
This recipe is for a true Texas chili, made with beef, chili peppers, and minimal additions. It's hearty, flavorful, and perfect for a chilly day.
**Ingredients:**

* 2 pounds ground beef
* 1 large onion, chopped
* 2 bell peppers (red or green), chopped
* 1 teaspoon chili powder
* 1 teaspoon cumin
* 1 teaspoon garlic powder
* 1/2 teaspoon cayenne pepper (or more, to taste)
* 1 (28 ounce) can crushed tomatoes
* 1 (15 ounce) can kidney beans, drained and rinsed
* Salt and pepper to taste

Figure 17: Example generations in the explicit, implicit, and steered setting for the prompt "Write me a recipe for a local dish." This is done with the American steering vector and explicit prompt.

**Explicit:** Hey! I was thinking about how we could use another reason to get together and celebrate, so I came up with an idea!
I call it **Luminescence Day**, celebrated on the first full moon of autumn.
Imagine: everyone gathering outdoors, in parks or backyards, with lanterns, fireflies in glass jars, or even glow sticks!
Think of it like a giant, peaceful, glowing picnic under the harvest moon. People would share stories and songs with the soft light illuminating their faces.
We could have this weird, cool tradition of writing down wishes on paper lanterns and releasing them into the sky, watching them drift up with the soft glow fading into the night.
Think of the Instagrammable moments!
There could be contests for the most creative lantern designs, or even workshops on how to make your own glowing decorations.
It'd be a really magical way to acknowledge the changing seasons and connect with each other under the beautiful autumn moon. You in?

**Implicit:** ## Global Story Sharing Day
**Date:** Last Saturday of August
**Concept:** Global Story Sharing Day is a day dedicated to celebrating the power of storytelling across cultures and generations. It encourages people to share stories – personal anecdotes, myths, legends, folk tales, jokes, poems – fostering connection and understanding.
**How People Celebrate:**

* **Storytelling Gatherings:** Communities organize events in parks, libraries, community centres, and even online platforms. Some gatherings focus on specific themes, like childhood memories, travel experiences, or local legends.
* **Open Mic Nights:** Public spaces host open mic nights where anyone can share their stories, creating a platform for diverse voices.
* **Sharing Circles:** Families and friends gather at home for intimate storytelling sessions, encouraging younger generations to hear and learn from the wisdom and experiences of their elders.
* **Digital Storytelling:** Platforms like social media, blogs, and YouTube become buzzing hubs for people to share their stories through written posts, videos, audio recordings, and even interactive storytelling experiences.
* **Story Swap Challenges:** Online challenges encourage people to swap stories with someone from a different culture or generation, expanding perspectives and creating global connections.
* **Storywriting Contests:** Competitions are held, inviting people to

**Steered:** ## National Gratitude Day: A Celebration of Appreciation
**Date:** The first Monday in November
**Concept:** National Gratitude Day is a holiday dedicated to expressing appreciation for the good things in life, big and small. It's a day to celebrate the blessings of family, friends, community, and even the simple pleasures of everyday life.
**Celebrations:**
* **Gratitude Walks:** Families and friends gather for walks, hikes, or bike rides, taking time to appreciate the beauty of their surroundings and the freedom to enjoy them.
* **Thank You Feasts:** Homes across the nation host feasts, potlucks, or barbecues, emphasizing shared meals and expressing gratitude to loved ones for their contributions.
* **Community Service Projects:** Many participate in volunteer activities, giving back to their communities and showing appreciation for the support they receive.
* **Gratitude Journals:** Individuals dedicate time to writing down all the things they are grateful for, reflecting on their blessings and fostering a sense of contentment.
* **"Thank You" Cards & Gifts:** People exchange thank you cards, expressing appreciation for friends, family, and even strangers who have impacted their lives positively. Gifts are often homemade or symbolic, representing a shared experience or a cherished memory

Figure 18: Example generations in the explicit, implicit, and steered setting for the prompt "Invent a new holiday and describe how people celebrate it." This is done with the American steering vector and explicit prompt.

**Turkey**: "Let's describe the **Dabke**, a traditional dance from the Levant region of the Middle East, including countries like Lebanon, Syria, Jordan, and Palestine.

*Here's a glimpse of the Dabke[...]
**France**: Let's describe the **Cancan**, a quintessential French dance!

**The Cancan:**

**Origin:** Born in the 1830s in Paris, it was a staple of the Parisian cabaret scene at the Moulin Rouge[...]
**Russia**: Let's describe the **Khorovod**, a traditional Russian dance:

**The Dance:**

**Formation:** The Khorovod is a circle dance, with dancers holding hands and moving in a circle around a central point, often[...]
**Bengladesh**: Let's describe the **Bharatanatyam**, a classical dance form from **Tamil Nadu, India**.

**Here's what makes it traditional:**

* **Ancient Roots:** Bharatanatyam is rooted in ancient Hindu temple rituals[...]
**United States**: Let's talk about the **square dance**, a quintessential American tradition!

**The Basics:**

* **Formation:** Square dancers form a square, with four couples facing each other.
* **Music:** Square dances are usually[...]

Figure 19: Anecdotal steering examples for prompt "Can you describe me a traditional dance?" with per-culture steering vectors applied on layer 25, $\alpha = 2$.

**Turkey**: "**Muhammara - Syrian Red Pepper and Walnut Dip**

**Muhammara** is a delicious and vibrant dip from Syria, made with roasted red peppers, walnuts, pomegranate molasses, and spices. It's traditionally served with[...]"
**France**: "Cassoulet Toulousain (Toulouse Cassoulet)

This is a classic dish from Toulouse, France, known for its rich flavors and hearty ingredients.

**Ingredients:**

* **For the Cassoulet:[...]"
**Russia**: "Borscht (Ukrainian Beetroot Soup) **This recipe is for a traditional Ukrainian borscht, a hearty and flavorful beetroot soup that is a staple in Ukrainian cuisine.**

**Ingredients:**

* **For the borscht:[...]"
**Bengladesh**: "Chana Masala (Bengali Chickpea Curry) **This is a flavorful and aromatic chickpea curry, a staple in Bengali cuisine. It's traditionally made with Bengal gram (chana), potatoes, and a blend of spices[...]"
**United States**: "New England Clam Chowder (Classic) This recipe is a classic New England staple, packed with clams, potatoes, and bacon. It's a hearty and comforting dish perfect for a chilly day. **Ingredients:[...]"

Figure 20: Anecdotal steering examples for prompt "Write me a recipe for a local dish." with per-culture steering vectors applied on layer 25, $\alpha = 2$.

## C  EXPLICIT–IMPLICIT LOCALIZATION GAP

We include Figure 21 to illustrate performance of each model by subtask.

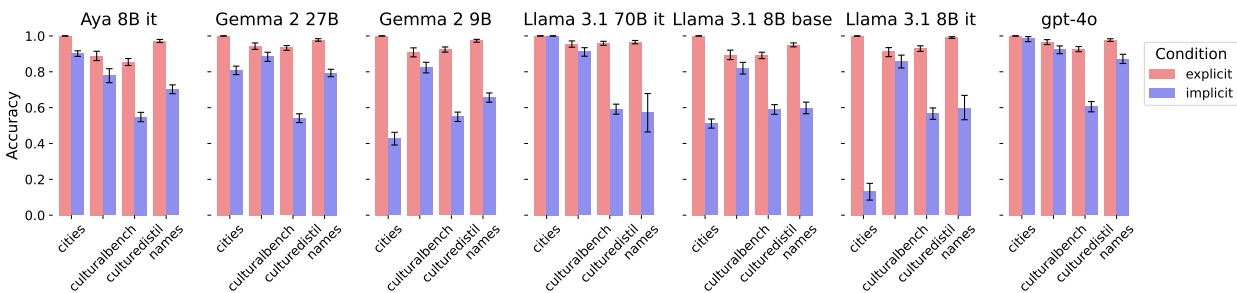

Figure 21: Performance of the different models by subtask in the explicit and implicit localization setting.

**Prefix analysis.** Examples of prefix words that are added to test what words lead to localization, relevant to Section 3.1.

- **France**: Baguette, Paris, Euro, Guillotine
- **Turkey**: Baklava, Istanbul, Lira, Nazar
- **USA**: Burger, New York, Dollar, Cowboy Hat
- **Bangladesh**: Biriyani, Dhaka, Taka, Rickshaw
- **Russia**: Borscht, Moscow, Ruble, Matryoshka

**Error analysis.** When no context is provided, some models have a tendency to refuse to provide the answer. In Table 10 we show the fraction of answers by model that fail to provide an answer.

| subtask | cities | culturalbench | o1-distilled | names |
|---|---|---|---|---|
| aya_8b_it | 0.125 | 0.169 | 0.113 | 0.183 |
| gemma2_27b_it | 0.000 | 0.000 | 0.000 | 0.000 |
| gemma2_9b_it | 0.004 | 0.009 | 0.000 | 0.001 |
| gpt4o | 0.638 | 0.044 | 0.086 | 0.319 |
| llama31_70b_it | 0.000 | 0.001 | 0.000 | 0.005 |
| llama31_8b_base | 0.000 | 0.000 | 0.000 | 0.000 |
| llama31_8b_it | 0.000 | 0.000 | 0.000 | 0.000 |

Table 10: Error rate with no context.

## D  STEERING PROMPTS AND PARAMETERS

First, we report the prompt format we used for defining the steering vectors since it deviated a little from the prompt used for activation patching. We added a "My guess is **" suffix at the end of the prompt to encourage generation, and removed numbering in the prompt options. This encouraged the model to generate the entire output of the question and empirically worked better for steering.

---

<bos><start_of_turn>user

Question:You must select one option and answer.  First, state the selected option in full, then explain your guess. What is a common living arrangement for children after they reach adulthood?

Options:

Children go to live with their distant relatives for better education or job opportunities.

Children often continue to live with their parents, or have their parents move into their homes to take care of them<end_of_turn>

<start_of_turn>model
My guess is **

---

Figure 22: Example prompt used for model steering.

**Hyperparameters for steering.** In Table 11, we show the best alpha and layer for steering the model.  The best parameter is determined by the task the steering vector was defined to solve.

- $v_{\text{tr.}}$: Best performance on localizing the translated prompts with no localization for each culture.
- $v_{\text{en}}$: Best performance on localizing the English prompts for each culture.
- $v_{\text{names}}$: Best performance on localizing the English prompts for each culture on the names subtask.
- $v_{\text{universal (tr.)}}$: Best performance on localizing the translated prompts with no localization for each culture in the held-one-out universal vector.
- $v_{\text{universal (en.)}}$: Best performance on localizing the English prompts for each culture in the held-one-out universal vector English.

|  | $v_{\text{tr.}}$ | | $v_{\text{en}}$ | | $v_{\text{names}}$ | | $v_{\text{universal (tr.)}}$ | |
|---|---|---|---|---|---|---|---|---|
|  | $l$ | $\alpha$ | $l$ | $\alpha$ | $l$ | $\alpha$ | $l$ | $\alpha$ |
| Bangladesh | 25 | 2 | 27 | 2 | 21 | 2 | 21 | 2 |
| France | 25 | 2 | 25 | 1 | 23 | 2 | 25 | 1 |
| Russia | 27 | 2 | 25 | 2 | 25 | 2 | 25 | 2 |
| Turkey | 21 | 2 | 24 | 2 | 25 | 2 | 27 | 2 |
| United States | 22 | 2 | 22 | 2 | 22 | 2 | 21 | -2 |

Table 11: Best-performing steering configuration per culture across all steering vectors.

**Implicit steering results.** In the main paper, we limited our analysis to steering on the explicit vector, however, we note that steering is also possible in the implicit context. The way we construct the implicit steering is by taking one translated prompt that correctly localized and subtracting its English variant. In Figure 23, we illustrate the steering performance across various alpha and layers. We see that steering leads to a minor improvement (on average around 7%), but still not close to the improvement from explicit steering.

**Effect of order position on steering.** One way of constructing the dataset would be to include two rows for each question with the question order changed. In Figure 24, we test to see the difference in steering performance with swapping options and not. Overall, we find that swapping the order of the questions has a negligible effect on steering vector performance. For this reason, we limit our analysis to non-swapped context for the steering analysis.

| $\alpha$ | Layer | Country | Implicit steering | Explicit steering | No context |
|---|---|---|---|---|---|
| 1 | 25 | Bangladesh | 0.696 | 0.739 | 0.628 |
| 2 | 26 | France | 0.595 | 0.612 | 0.547 |
| 1 | 25 | Russia | 0.614 | 0.793 | 0.532 |
| 2 | 23 | Turkey | 0.654 | 0.763 | 0.559 |
| 1 | 25 | United States | 0.621 | 0.695 | 0.518 |

Table 12: Steering results on the implicit questions. The $\alpha$ and layer show the best layer and alpha parameter for the implicit steering. Implicit steering column shows performance on cultural tasks when the implicit vector is added to the translated question. No context is baseline implicit performance. And explicit steering shows performance when explicit steering.

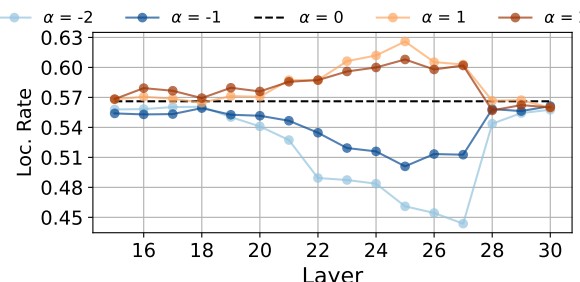

Figure 23: Steering results for per-culture vectors calculated in implicit setting on translated questions with $\alpha \in [-2, 2]$ across layers [15-30], where the x-axis represents the layer at which steering vector is applied, and the y-axis indicates the ratio of localized responses. For United States, localization performance is calculated in the reverse direction (i.e. English minus translated)

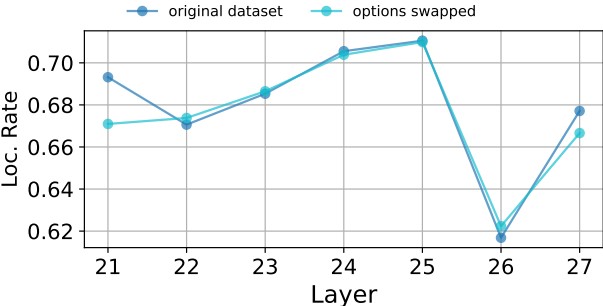

Figure 24: Steering results on the original and option-swapped datasets for per-culture vectors calculated using English pairs with $\alpha = 2$ across layers [21-27]. The x-axis represents the layer where the steering vector is applied, and the y-axis indicates the ratio of localized responses.

# E  MULTIPLE CHOICE QUESTION

## E.1  STEERING

Throughout our analysis we focused on the binary setting where one option was always from the West for simplicity. This is a problem for the steering analysis, since the steering vector may simply learn to answer the non-American answer. If we imagine a prompt where two of the options are non-American, the model then may not give the answer related to the language of the prompt. We now extend to the five-choice setting where multiple options are non-American. To do this we augment our prior datasets with more options. Since the questions are the same across languages, we add five randomized options from each culture for these datasets. But since the o1-distill data was culture-specific, we used o3-mini-high to regenerate a set of culturally relevant questions with one option for each of our five contexts, in a way similar to the o1-preview approach previously done. In this setting we evaluate both the translated steering vector and the universal translated steering vector.

In Table 13 we show the performance of the steering vector on the multiple choice alongside the performance of the culture-specific steering vector (explicit translated subtract implicit translated) and implicit/explicit baselines. We observe that the universal steering vector gets similar results to the culture-specific steering, but still lags behind the explicit setting.

| | $v_{\text{universal (tr)}}$ | $v_{\text{tr.}}$ | Implicit | Explicit |
|---|---|---|---|---|
| Bangladesh | 0.537 | 0.719 | 0.418 | 0.863 |
| France | 0.292 | 0.308 | 0.233 | 0.860 |
| Russia | 0.565 | 0.542 | 0.286 | 0.914 |
| Turkey | 0.441 | 0.526 | 0.285 | 0.841 |
| United States | 0.371 | 0.596 | 0.219 | 0.848 |

Table 13: Steering results for the universal (translated) and culture-specific (translated) steering vectors.

| | $v_{\text{tr.}}$ | | $v_{\text{universal (tr.)}}$ | |
|---|---|---|---|---|
| | $l$ | $\alpha$ | $l$ | $\alpha$ |
| Bangladesh | 24 | 2 | 21 | 2 |
| France | 27 | 2 | 22 | 2 |
| Russia | 23 | 2 | 23 | 2 |
| Turkey | 27 | 2 | 27 | 2 |
| United States | 25 | 2 | 22 | 2 |

Table 14: Optimal steering parameters for the multiple choice question answering task. The optimal parameters were done from a sweep of $\alpha \in [-2, -1, 0, 1, 2]$ and $l \in [21, ..., 27]$

# F    ACTIVATION PATCHING

## F.1    EXPLICIT (ENGLISH)

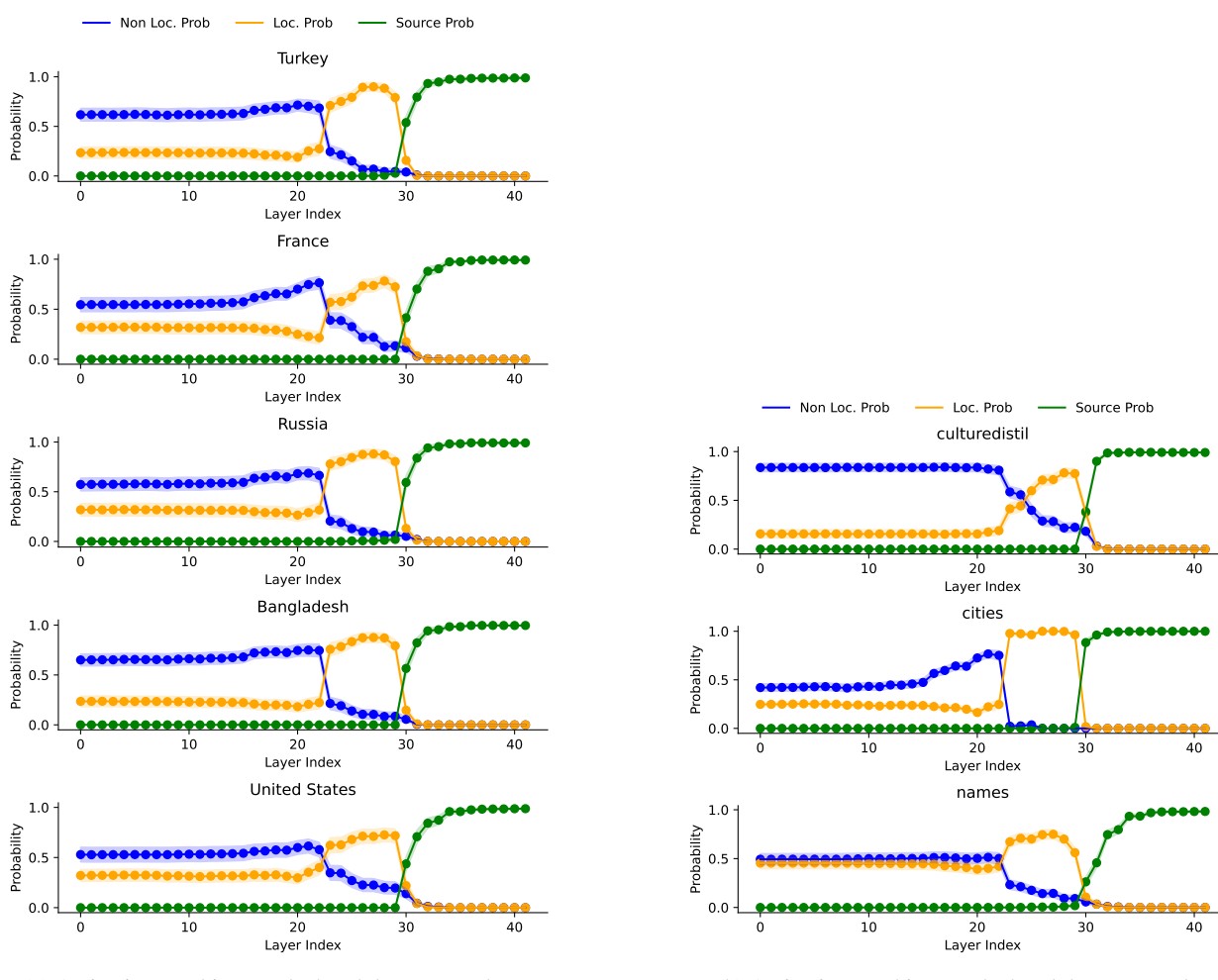

(a) Activation patching results breakdown per culture.

(b) Activation patching results breakdown per task.

Figure 25: Comparison of activation patching results per culture and per task. Target prompt localized token probability (Loc. Prob) is shown in yellow, and non-localized target prompt token probability (Non Loc. Prob) is shown in blue. Green shows the probability of answering the question from the source prompt. Shaded regions represent 95% confidence intervals (CI) as mean $\pm 1.96 \times$ SEM. Source-English prompt with context and target English prompt.

## F.2 EXPLICIT (TRANSLATED)

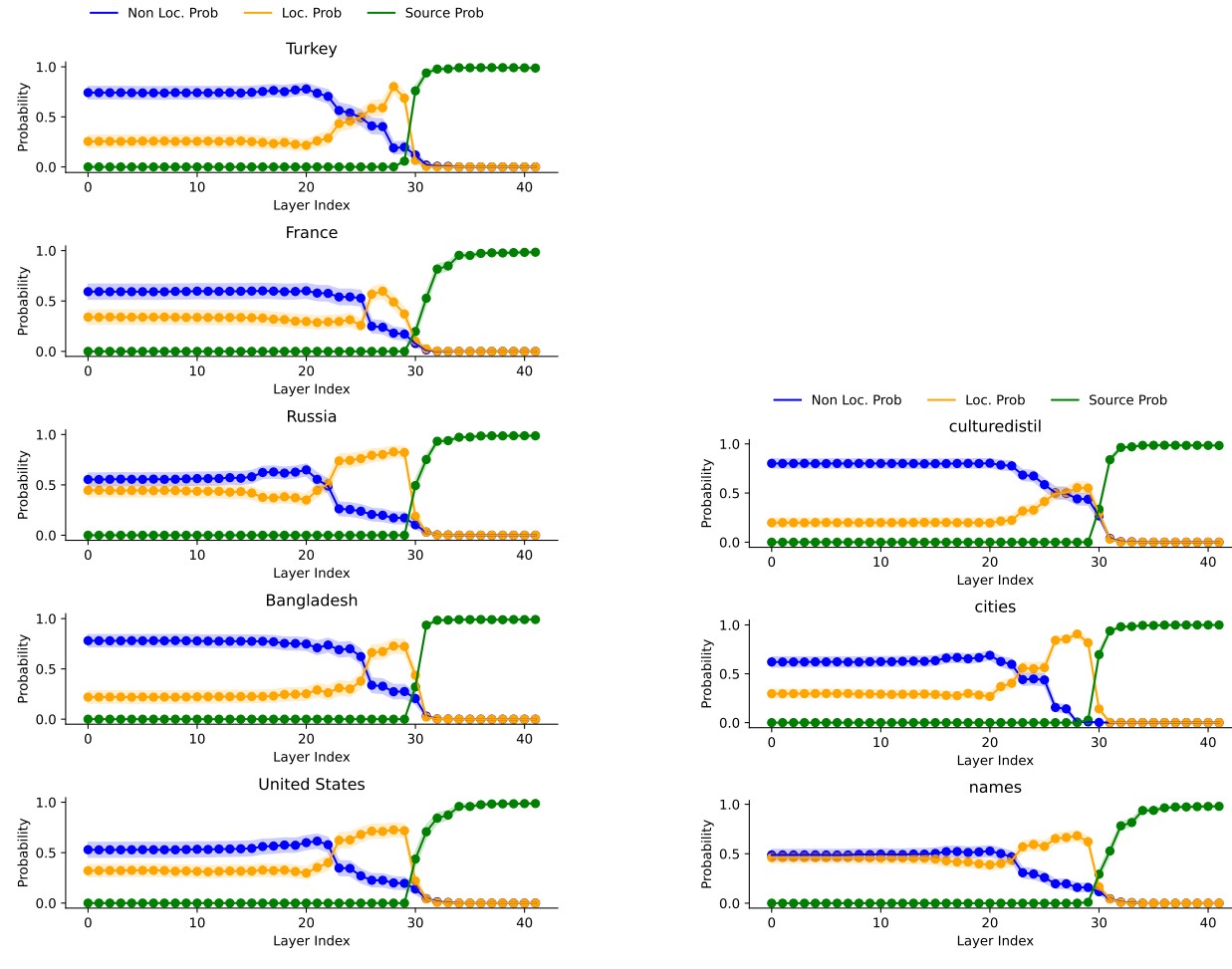

(a) Activation patching results breakdown per culture.

(b) Activation patching results breakdown per task.

Figure 26: Comparison of activation patching results per culture and per task. Target prompt localized token probability (Loc. Prob) is shown in yellow, and non-localized target prompt token probability (Non Loc. Prob) is shown in blue. Green shows the probability of answering the question from the source prompt. Shaded regions represent 95% confidence intervals (CI) as mean $\pm 1.96 \times$ SEM.

## F.3 IMPLICIT

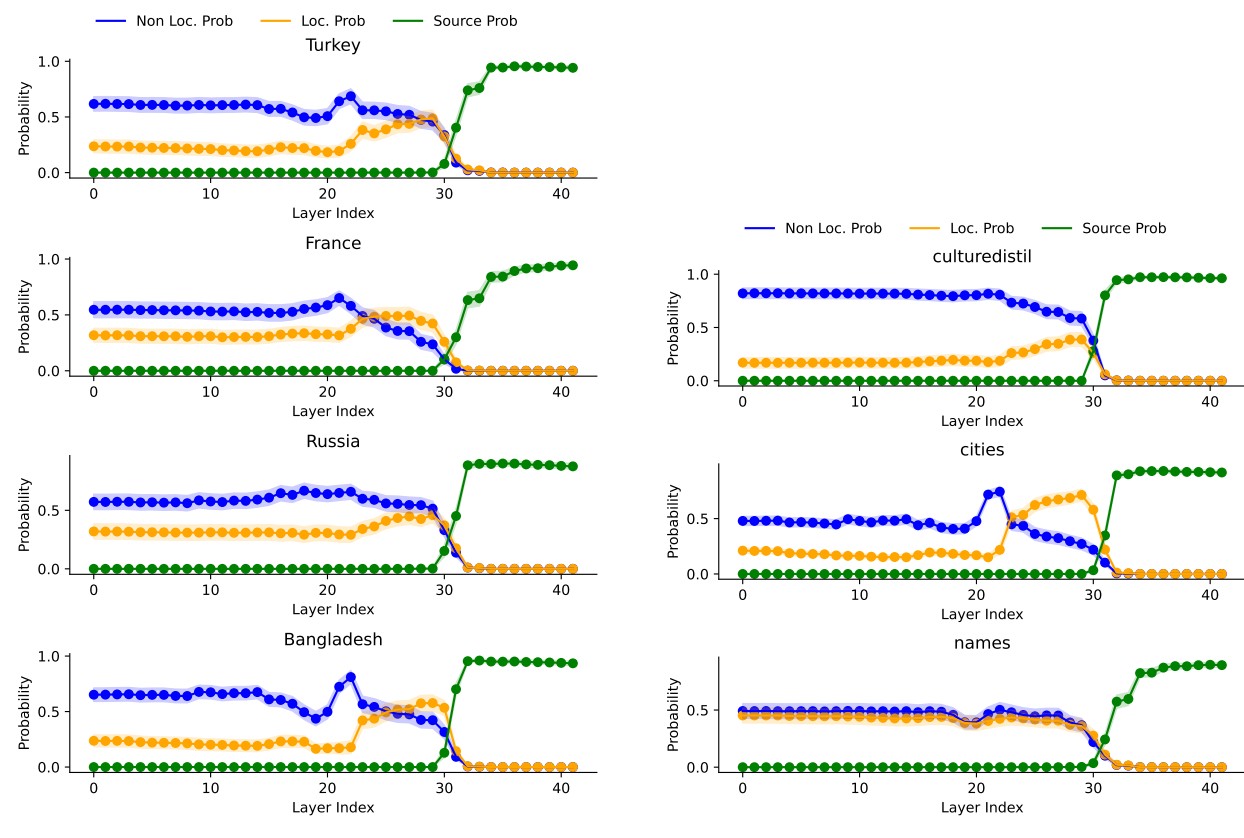

(a) Activation patching results breakdown per culture.

(b) Activation patching results breakdown per task.

Figure 27: Comparison of activation patching results per culture and per task. Target prompt localized token probability (Loc. Prob) is shown in yellow, and non-localized target prompt token probability (Non Loc. Prob) is shown in blue. Green shows the probability of answering the question from the source prompt. Shaded regions represent 95% confidence intervals (CI) as mean $\pm\ 1.96 \times$ SEM. Source-translated prompt and target English prompt.

