# OpenReview forum: "Localized Cultural Knowledge is Conserved and Controllable in Large Language Models"
_ICLR.cc/2026/Conference — ICLR 2026 Conference Withdrawn Submission_

### Official Review · Reviewer_LYar · 2025-10-21

**Soundness:** 3
**Presentation:** 3
**Contribution:** 2
**Rating:** 4
**Confidence:** 4

**Summary:**

This paper investigates where large language models (LLMs) store local cultural information by examining their internal representations. It first explores the effects of explicit cultural context prompts (i.e., providing explicit country information) and implicit cultural context prompts (i.e., using multilingual input), demonstrating that cultural knowledge requires explicit context to surface in multilingual settings. The paper then identifies an explicit cultural customization vector that steers LLMs toward non-English cultures. Finally, it shows the generality of these findings across seven models and four cultures.

**Strengths:**

S1: The paper investigates an interesting and timely problem of how to steer language models toward specific cultures. The framing of the implicit vs. explicit localization gap provides a nice way to conceptualize differences in model behaviour.

S2: The paper presents an extensive investigation of the localization gap, layer-wise importance, and steering experiments, using both synthetic and real datasets.

S3: Comparing localization and cultural adaptation through steering vectors (instead of prompting) is a good approach that goes beyond black-box prompting. The hypothesis of universal steering vectors is particularly interesting and shows strong potential.

**Weaknesses:**

W1: There are several potential evaluation concerns, including:

- Lack of human evaluation: The use of cosine similarity between embeddings serves only as a proxy for semantic similarity, rather than a direct human judgment of cultural appropriateness. The paper would be stronger if human raters from the target cultures were involved in evaluating the open-ended generations, especially given the relatively small number of prompts used (24, L186).
- Using GPT embeddings with GPT as the judge: This raises concerns about circularity and potential bias in evaluation.

W2: While the hypothesis of a universal cultural steering vector is enticing, the supporting evidence is relatively limited. More ablations (how robust across tasks, sparsity, what negative side-effects occur) would strengthen the claim. Extending the evaluation to include more low-resource languages would also enhance the findings. Finally, expanding the study to cases where cultures differ but the language is roughly the same (e.g., the U.S., U.K., Australia, and Singapore) would further strengthen the work.

W3: Despite using a definition of culture in L078, the concept of culture remains vaguely *operationalized* in the chosen evaluation data. It is unclear which aspects of culture each evaluation dataset examines, or how these collectively provide a comprehensive view of the intended test.

**Questions:**

- Are authors planning on releasing all the synthetic data for reproducibility?
- Did you examine whether the steering vector introduces any negative side effects on unrelated tasks, such as factual correctness or language fluency?
- In general, how should one choose which layer to steer for universal steering?

Other comments:
- L087 - citation should be added the first time mentioning CulturalBench.
- While Aya-8B is one explicitly multilingual model, Aya-8B-Expanse is a much stronger variant. I am very curious to see whether the results and findings also hold for Aya-8B-Expanse.

---

> ### Author Response · Authors · 2025-11-24
> **Response to reviewer**
>
> We want to thank the reviewer for taking the time to review our paper. We hope that we address some of the concerns / questions of the reviewer below:
> W1: Lack of human evaluation: The use of cosine similarity between embeddings serves only as a proxy for semantic similarity, rather than a direct human judgment of cultural appropriateness. The paper would be stronger if human raters from the target cultures were involved in evaluating the open-ended generations, especially given the relatively small number of prompts used (24, L186).
> R1: We thank the reviewer for making this point. We want to clarify where we use the embeddings. The embeddings are only used when measuring homogeneity (Table 1) and not used for measuring the cultural appropriateness of the generations. Instead, we rely on GPT-as-a-judge for measuring the cultural faithfulness and stereotypicality. We now include a reasoning model (GPT OSS 120B) to add another measure of stereotypicality/faithfulness. We include a table here showing the explicit win-rate for both GPT-OSS and GPT-4o. We note that there is consistency across both models.
> | Model | gpt-4o | gpt-oss-120b |
> |:------|:-------|:-------------|
> | gemma-2-9b | 0.8863 | 0.8586 |
> | gemma-2-27b | 0.8714 | 0.8589 |
> | gpt-4o | 0.8711 | 0.7877 |
> | llama-3.1-70b-instruct-turbo | 0.7958 | 0.8557 |
> | llama-3.1-8b-instruct-turbo | 0.8657 | 0.8575 |
>
> W2: Using GPT embeddings with GPT as the judge: This raises concerns about circularity and potential bias in evaluation.
>
> We want to note that the GPT-as-a-judge and GPT embeddings never interact. The embeddings are used only when compared with each other for measuring the diversity.
>
> W3: While the hypothesis of a universal cultural steering vector is enticing, the supporting evidence is relatively limited. More ablations (how robust across tasks, sparsity, what negative side-effects occur) would strengthen the claim.
>
> R3: We completely agree. The universal steering vector is worthy of a paper in-and-of-itself but not a main contribution of this paper.
>
> To reiterate: in this paper we focused on studying if language alone is enough to culturally localize a generation. In other words, if someone from language X asks the model to generate an answer it will not always localize to the culture to the people who speak that language. Adding explicit context, (like “I am from country X”) makes the model almost perfect at localizing the answer, but leads to more stereotypical and homogenous responses.
>
> We next study the mechanisms behind cultural localization and find that adding a linear steering vector leads to good performance; without the downsides of explicit prompting. The universal steering hypothesis was a final analysis that we added and is a motivation for future research.
>
> W4: Extending the evaluation to include more low-resource languages would also enhance the findings. Finally, expanding the study to cases where cultures differ but the language is roughly the same (e.g., the U.S., U.K., Australia, and Singapore) would further strengthen the work.
>
> R4: We completely agree and hope that future work can expand on our findings here.

---

> ### Author Response · Authors · 2025-11-24
> **Response to reviewer**
>
> W5: Despite using a definition of culture in L078, the concept of culture remains vaguely operationalized in the chosen evaluation data. It is unclear which aspects of culture each evaluation dataset examines, or how these collectively provide a comprehensive view of the intended test.
>
> R5: We thank the reviewer for pointing this out. This is something we ourselves wrestled with but with this external feedback we’ve sharpened our exploration in a couple of ways:
>
> We now more explicitly note that: Our benchmark does not require that a language corresponds to a single country; rather, our evaluation relies only on relative frequency differences between the culturally local option and the U.S. option within the dominant cultural distribution associated with that language. In the Bengali case, this requires only that the local name be more likely than the American option—not that it be universally common among all Bengali speakers. We recognize that this nuance was not sufficiently clear in the manuscript, and we now clarify this assumption and explicitly incorporate it into the limitations section.
>
> We also note that we’ve added another analysis. In the explicit setting before we added the prefix “I live in Bangalore". But now we adjusted the explicit setting to also have “I speak Bengali”. We observe that in this case the explicit-implicit localization gap is the same. We included the table in responses but also including it here.
>
> We’ve made the connection with culture to be less pronounced and focus more on having the model answer a prompt in “Bengali” the same way that if we explicitly add “I speak Bengali to the prompt”.
>
> | model             | setting       |     mean | 95% CI         |
> |:------------------|:--------------|---------:|:---------------|
> | Aya Expanse 8B    | explicit      | 0.76259  | [0.733, 0.793] |
> | Aya Expanse 8B    | implicit      | 0.696932 | [0.681, 0.713] |
> | Aya Expanse 8B    | language hint | 0.83369  | [0.822, 0.846] |
> | Gemma 2 27B       | explicit      | 0.878004 | [0.859, 0.896] |
> | Gemma 2 27B       | implicit      | 0.72666  | [0.714, 0.739] |
> | Gemma 2 27B       | language hint | 0.881456 | [0.873, 0.891] |
> | Gemma 2 9B        | explicit      | 0.881356 | [0.862, 0.899] |
> | Gemma 2 9B        | implicit      | 0.603684 | [0.588, 0.620] |
> | Gemma 2 9B        | language hint | 0.838022 | [0.827, 0.848] |
> | Llama 3.1 70B it  | explicit      | 0.879713 | [0.859, 0.898] |
> | Llama 3.1 70B it  | implicit      | 0.739053 | [0.721, 0.758] |
> | Llama 3.1 70B it  | language hint | 0.871773 | [0.862, 0.881] |
> | Llama 3.1 8B base | explicit      | 0.849421 | [0.829, 0.871] |
> | Llama 3.1 8B base | implicit      | 0.543255 | [0.528, 0.560] |
> | Llama 3.1 8B base | language hint | 0.605436 | [0.592, 0.619] |
> | Llama 3.1 8B it   | explicit      | 0.898488 | [0.879, 0.917] |
> | Llama 3.1 8B it   | implicit      | 0.554557 | [0.530, 0.580] |
> | Llama 3.1 8B it   | language hint | 0.64545  | [0.633, 0.661] |
> | gpt-4o            | explicit      | 0.891791 | [0.873, 0.910] |
> | gpt-4o            | implicit      | 0.775908 | [0.758, 0.793] |
> | gpt-4o            | language hint | 0.885327 | [0.877, 0.894] |
>
>
> Questions:
> Q1: Are authors planning on releasing all the synthetic data for reproducibility?
> - Yes! The repository with all data will be made public upon acceptance.
>
> Q2: Did you examine whether the steering vector introduces any negative side effects on unrelated tasks, such as factual correctness or language fluency?
> - Thank you for this. The only analysis that we ran using the steering analysis was faithfulness / stereotypicality (Table 7 and 8 in paper). But you’re right, maybe there are more subtle ways for how the steering results can affect model generations. To account for this, we now run an additional fluency test that we report here and include to the Appendix. We note that steering didn’t lead to a drop in fluency, but was actually preferred by the models. The prompt used for getting these fluency scores is included in Figure 13 in the paper.
>
> | Language | Explicit               | Steering               |
> |----------|-------------------------|-------------------------|
> | Bengali       | 0.108 ± 0.016          | 0.886 ± 0.017          |
> | Russian       | 0.269 ± 0.023          | 0.667 ± 0.026          |
> | French       | 0.361 ± 0.023          | 0.492 ± 0.025          |
> | English       | 0.283 ± 0.024          | 0.508 ± 0.026          |
> | Turkish       | 0.269 ± 0.023          | 0.706 ± 0.023          |

---

> ### Author Response · Authors · 2025-11-24
> **Response to reviewer**
>
> Q3: In general, how should one choose which layer to steer for universal steering?
> - To do this we have Figure 22 and 23. We selected the layer that led to the greatest reduction in the E-I localization gap on a train set. We note that similar to past work [1] steering the language model around layer 20 begins to lead to increased localization performance.
>
> Other comments:
>
> L087 - citation should be added the first time mentioning CulturalBench.
>
> - Thank you for mentioning this. We added this.
>
> While Aya-8B is one explicitly multilingual model, Aya-8B-Expanse is a much stronger variant. I am very curious to see whether the results and findings also hold for Aya-8B-Expanse.
> - We actually do test Aya-8B-Expanse. We now make it more clear in the text. However, we fully recognize where this confusion came from since we cited the wrong paper when introducing the model.
>
>
> [1] Dumas, C., Wendler, C., Veselovsky, V., Monea, G., & West, R. (2025, July). Separating tongue from thought: Activation patching reveals language-agnostic concept representations in transformers. In Proceedings of the 63rd Annual Meeting of the Association for Computational Linguistics (Volume 1: Long Papers) (pp. 31822-31841).

---

> > ### Comment · Reviewer_LYar · 2025-11-25
> >
> > Thank you very much for the responses and additional clarifications.
> > While some of my concerns have been clarified, others remain largely unaddressed. For example, I am not convinced that using another GPT model (gpt-oss-120b) as a judge can replace human judgment on cultural faithfulness and stereotypes, nor does it provide substantial additional benefit compared to GPT-4o, which is already used in the current paper.
> >
> > I think the paper would benefit from a revision with better design experimental setup. Hence I maintain my scores as they reflect my current assessment.

---

### Official Review · Reviewer_sbiU · 2025-10-24

**Soundness:** 2
**Presentation:** 1
**Contribution:** 2
**Rating:** 2
**Confidence:** 4

**Summary:**

The paper investigates cultural localization in large language models, meaning to identify which internal layers encode culture-specific information and how to steer model behavior toward certain cultural traits. The authors introduce a cultural localization benchmark with four datasets and a method to locate culturally sensitive layers using activation patching. They propose benchmarks based on names and city-related questions and evaluate explicit vs. implicit cultural elicitation, as well as open-ended generation tasks. The study’s main claim is that large models contain latent cultural knowledge that can be mechanically accessed.

**Strengths:**

- The attempt to systematically localize cultural traits within model layers and define a linear “steering” mechanism is a technically interesting direction. Understanding and controlling culturally grounded behaviors is important for fairness and model transparency.

- The proposed framework could open avenues for better evaluation of cultural competence and for designing more adaptive models.

- The overall motivation and broader implications are clearly articulated in the introduction and conclusion.

**Weaknesses:**

- The paper is hard to follow. Dataset descriptions are confusing. Several key sentences are grammatically incorrect or incomplete (e.g. "Mathematically, the steering vector at layer j steers away from the behavior displayed by negative examples from D− and towards the behavior of positive examples D+ is defined as", "If the average similarity is higher than answers we consider those answers more semantically similar.")

- The procedure for activation patching and steering is not sufficiently explained. It is unclear what _last token position_ $t_{source}$ refers to, which tokens are replaced in what layers, or how source-target pairs are constructed. Likewise, “positive” and “negative” examples ($D^+$ and $D^-$) are never clearly described.

- The data labels of “correct English answer” or “correct translated answer” in the explicit–implicit localization gap are not defined and thus confusing. As a result, Figure 1 and some quantitative results cannot be interpreted.

- The cosine similarity measure between text embeddings of different answers are not informative of homogeneity, the interpretation of values (e.g., 0.333 vs. 0.359) is unclear, in what sense is different?  the words? the topic? Also, reliance on ChatGPT as a cultural judge is questionable.

- Reported results (e.g., 64% and 32% refusal rates from GPT-4o to answer city and name-related questions) are difficult to interpret without clear dataset sizes.

- The central claim that the method finds layers where “cultural localization happens” is intriguing but difficult to understand due to the lack of clarity in the methodology.

**Questions:**

1. Please clarify the activation patching procedure: what are $t_{source}$ and $t_{target}$? Which tokens are replaced in what layers, and how are the source–target pairs selected from the dataset?

2. Please clarify the datasets. How do you define “positive” and “negative” examples when constructing $D^+$ and $D^-$ for the steering vector? What does it mean for a response to be “correct” or “incorrect” in the explicit–implicit localization task? Could you provide dataset statistics (number of samples per culture, domain, etc.) to assess the robustness of your results?

3. How to interpret the cosine similarity score? Two pairs of sentences could have the same similarity score for two very different reasons.   How should we interpret it in semantic terms?

---

> ### Author Response · Authors · 2025-11-24
> **Response to reviewer**
>
> Thank you for taking the time to thoroughly reviewing our work, we appreciate your constructive feedback and will make sure to use it to enhance the clarity of our paper. We proceed by addressing your weaknesses and questions raised one by one.
>
> W1. The paper is hard to follow. Dataset descriptions are confusing. Several key sentences are grammatically incorrect or incomplete (e.g. "Mathematically, the steering vector at layer j steers away from the behavior displayed by negative examples from D− and towards the behavior of positive examples D+ is defined as", "If the average similarity is higher than answers we consider those answers more semantically similar.")
>
> R1. Thank you for pointing out the grammatical errors. We’ve fixed them. We are confident that our responses to the remaining more targeted clarification requests will enhance the clarity of our work and make it easier to follow.
>
> W2. The procedure for activation patching and steering is not sufficiently explained. It is unclear what last token position  refers to, which tokens are replaced in what layers, or how source-target pairs are constructed. Likewise, “positive” and “negative” examples ( and ) are never clearly described.
>
> R2. Thank you for the comment. We will reiterate and slightly expand the explanation here for clarity.
> In line 125, when we define our input as a sequence of tokens $(x_1, \dots, x_t)$, the last token is $x_t$ and the position of the last token is $t$; this is what we refer to as the last token position. In the context of activation patching, there is a source sequence $x_{\text{source}}$ made up of tokens $x_{\text{source},1}, \dots, x_{\text{source}, t_{\text{source}}}$ and a target sequence $x_{\text{target}}$ made up of tokens $x_{\text{target},1}, \dots, x_{\text{target}, t_{\text{target}}}$ (introduced in line 146). In our paper, we only consider patching between representations computed at the last token positions $t_{\text{source}}$ and $t_{\text{target}}$, and we only patch within the same layer (indicated by the superscript ${}^{(j)}$ in line 153). The notation that we use to refer to the activations (e.g., $h_t^{(j)}$) is introduced in line 130.
>
> Thus, to the best of our understanding, “last token position” and “which tokens are replaced in what layers” are defined in the relevant part of our paper. However, to enhance clarity, we will explicitly expand $x_{\text{source}}$ and $x_{\text{target}}$ into their constituent token sequences as done above. Please let us know if you consider this sufficient or if there is anything else we could improve about the presentation.
>
> “How are source–target pairs constructed?” For activation patching, we introduce the concrete source and target pairs that we use in Section 3.2 (lines 266–299). We provide concrete examples of source–target pairs in lines 269 and 298. For convenience, we summarize them here:
>
> **Implicit localization setting.**
>
>  We take an English question and its direct translation into the target language.
>
>  “[...] Concretely, the English version may look like ‘A common name here is (1) George, (2) Sergey. Answer:’ where the model would answer George. The Russian copy would be a mere translation of this. [...]”
> In this setting, we use the translated version as the source prompt and the English version as the target prompt.
>
> **Explicit localization setting.**
>
>  “[...] we use only translated examples [...] Here, we pair target prompts without context with their counterparts including explicit cultural context as source prompts [...].”
>
>  For example, the cultural context could be “I live in the United States” or “I live in Russia”.
>
> In this setting, we use the translated version with cultural context as the source prompt and the translated version without cultural context as the target prompt.
>
> Example for the explicit case:
>
> Source prompt:
>
> ```
> <bos><start_of_turn>user
> Question: To answer the following multiple-choice question, you should choose one option only among A,B. Instruction: You must select one option among A,B. Do not output any other things.
> I live in the United States. Which of the following names is most likely to be my name?
> A. Liam
> B. Alparslan
> <start_of_turn>model\n
> ```
>
> Target prompt:
>
> ```
> <bos><start_of_turn>user
> Question: To answer the following multiple-choice question, you should choose one option only among 1,2. Instruction: You must select one option among 1,2. Do not output any other things.
> Which of the following names is most likely to be my name?
> 1. Liam
> 2. Alparslan
> <start_of_turn>model\n
> ```
>
> [CONTINUED IN NEXT MESSAGE]

---

> > ### Author Response · Authors · 2025-11-24
> > **Response to reviewer**
> >
> > Here, we patch at the last token of the prompt, which is “\n”, and patching is applied at each layer. The x-axis in Fig. 3 denotes the layer at which patching is performed.
> >
> > In this example, the localized answer is Liam, i.e., option “1”. Accordingly, in Fig. 3 we plot:
> >
> > $\text{Loc. Prob.} = P(y = 1 \mid x_q)$,
> >
> >
> > $\text{Non Loc. Prob.} = P(y = 2 \mid x_q)$, and
> >
> >
> > $\text{Source Prob.} = P(y = a_{\text{source}} \mid x_q)$,
> >
> >
> > where $a_{\text{source}} \in {A,B}$ is the option selected by the model for the source prompt before patching, and $(x_q)$ denotes the patched representation at the last token position. All probabilities are computed via a softmax with temperature (1). We will clarify these three quantities in the caption and text for Fig. 3.
> >
> > Similarly, for steering, we introduce the high-level construction of the datasets $D^+$ and $D^-$ in Section 3.3 (lines 321–323) and refer to Appendix D for the concrete prompt template.
> >
> > W3: The data labels of “correct English answer” or “correct translated answer” in the explicit–implicit localization gap are not defined and thus confusing. As a result, Figure 1 and some quantitative results cannot be interpreted.
> >
> > R3: Thank you for pointing out this ambiguity,  we agree that the data labels need to be clarified.
> > Each question in our dataset is constructed with exactly one culturally localized option (appropriate for the target culture) and one non-localized/irrelevant option. We call a generation correct if the model selects this culturally localized option for the culture implied by the prompt language (implicit setting) or explicitly mentioned in the prompt (explicit setting); otherwise it is incorrect.
> > In the definition of the explicit–implicit localization gap, the two labels are:
> > $y_{\text{loc}}$ (“correct English answer”): the culturally localized option when the question is presented in English, with explicit context $x_{\text{context}}$ (e.g., “I live in Russia”).
> >
> >
> > $y_{\text{tr. loc}}$ (“correct translated answer”): the same culturally localized option, but in the translated version of the question $x_{\text{tr. } q}$, written in the target language.
> >
> >
> > For the United States, the English and “translated” versions coincide, so $(y_{\text{loc}})$ and $(y_{\text{tr. loc}})$ are identical, but we keep the notation for consistency across cultures.
> >
> > We will revise the paragraph around the EI-Gap definition (and the caption of Figure 1) to make these label definitions explicit.
> >
> > W4: The cosine similarity measure between text embeddings of different answers are not informative of homogeneity, the interpretation of values (e.g., 0.333 vs. 0.359) is unclear, in what sense is different? the words? the topic? Also, reliance on ChatGPT as a cultural judge is questionable.
> >
> > R4: Thank you for this comment. We want to first point to the fact that using text embeddings as a measure of textual similarity is frequently done in the literature as a measure of semantic diversity [1,2,3]. We provide more context to your question below.
> >
> > [1] Zhang, T., Peng, B., & Bollegala, D. (2024). Improving diversity of commonsense generation by large language models via in-context learning. arXiv preprint arXiv:2404.16807.
> >
> > [2] Zhu, A., Asawa, P., Davis, J. Q., Chen, L., Hanin, B., Stoica, I., ... & Zaharia, M. (2025). BARE: Leveraging Base Language Models for Few-Shot Synthetic Data Generation. arXiv preprint arXiv:2502.01697.
> >
> > [3] Shaib, C., Barrow, J., Sun, J., Siu, A. F., Wallace, B. C., & Nenkova, A. (2024). Standardizing the measurement of text diversity: A tool and a comparative analysis of scores. arXiv preprint arXiv:2403.00553.

---

> > > ### Author Response · Authors · 2025-11-24
> > > **Response to reviewer**
> > >
> > > W5: Reported results (e.g., 64% and 32% refusal rates from GPT-4o to answer city and name-related questions) are difficult to interpret without clear dataset sizes.
> > >
> > > R5: Thank you for this request for clarification. We have information on the dataset size in Table 5 (Appendix A.2). For convenience, we give the same table below.
> > >
> > > | **Subtask**     |  **BD** |  **FR** |  **RU** |  **TR** |  **US** | **Total** |
> > > |-----------------|--------:|--------:|--------:|--------:|--------:|----------:|
> > > | Cities          |     540 |     700 |     700 |     700 |     740 |     3380  |
> > > | CulturalBench   |     300 |     156 |     336 |     456 |     120 |     1368  |
> > > | CultureDistil   |     624 |     616 |     632 |     808 |     692 |     3372  |
> > > | Names           |     848 |     848 |     552 |     836 |     770 |     3854  |
> > > | **Total**       |   2312  |   2320  |   2220  |   2800  |   2322  |   11974   |
> > >
> > > W6: The central claim that the method finds layers where “cultural localization happens” is intriguing but difficult to understand due to the lack of clarity in the methodology.
> > >
> > > R6: We hope that with our clarifications, the central claim is now clearer. In our experiments, we operationalize “where cultural localization happens” as the layers whose activations have the strongest effect on whether the model outputs the culturally localized vs. non-localized answer. Concretely, we identify these layers as those where (i) activation patching between localized and non-localized prompts most strongly shifts the output, and (ii) applying the learned steering vector most reliably flips the model toward the localized answer. These two interventions consistently point to the same middle–to–late layers (23-27)
> > >
> > > Q1: Please clarify the activation patching procedure: what are  and ? Which tokens are replaced in what layers, and how are the source–target pairs selected from the dataset?
> > >
> > > A1: Answered in R2.
> > >
> > > Q2: Please clarify the datasets. How do you define “positive” and “negative” examples when constructing  and  for the steering vector? What does it mean for a response to be “correct” or “incorrect” in the explicit–implicit localization task? Could you provide dataset statistics (number of samples per culture, domain, etc.) to assess the robustness of your results?
> > >
> > > A2: For the explicit–implicit localization task, each question is constructed with exactly one culturally localized option (appropriate for the target culture) and one non-localized/irrelevant option. We call a generation correct if the model selects the culturally localized option for the culture implied by the prompt language (in the implicit setting) or the culture mentioned in the prompt (in the explicit setting); otherwise it is incorrect.
> > >
> > > When constructing the positive and negative sets for the steering vector, we proceed as follows. In the explicit setting, a positive example is a “hinted” prompt for which the model chooses the culturally localized option, and the corresponding negative example is the same question in a plain (non-hinted) form where the model instead chooses the non-localized option. In the implicit setting, a positive example is a translated prompt (into the target language) where the model’s answer is culturally localized, and the negative example is the same question in English where the model’s answer is non-localized. We will make these definitions explicit in the main text.
> > >
> > > Q3: How to interpret the cosine similarity score? Two pairs of sentences could have the same similarity score for two very different reasons. How should we interpret it in semantic terms?
> > >
> > > A3: Cosine similarity here measures how close two generations are in semantic space, not how similar their wording is. A higher score means the generations talk about roughly the same things; a lower score means they diverge in meaning. It’s not perfect – different pairs can land on the same score for different reasons – but it works well as a broad measure of whether generations collapse toward the same ideas or stay diverse.

---

> > > > ### Comment · Reviewer_sbiU · 2025-11-26
> > > >
> > > > Please keep in mind that, while you are very familiar with the notation and terminology because you worked on this project for a long time, a reader encountering the work for the first time does not. As it stands, the presentation in Section 2 (Materials and Methods) does not sufficiently guide the reader through the structure of the set up. I encourage you to reorganize this section with a clearer narrative to avoid confusion about names, references, and the meaning and difference of each thing.
> > > >
> > > > Regarding the cosine similarity measure between text embeddings: I am not questioning what cosine similarity is, nor whether it is commonly used as a proxy for semantic similarity. I fully understand those points. My concern is different: How meaningful are the specific differences you are reporting? Stating that cosine similarity “measures how close two generations are in semantic space” does not answer this. The question is about the relevance of the differences you observe, and how one should interpret these values in practice for the phenomena you study.
> > > >
> > > > In line with concerns raised by other reviewers, I also remain unconvinced that using GPT model as a cultural judge yields reliable measurements; the model’s own biases and limitations make its judgments insufficiently trustworthy for supporting any claims.

---

### Official Review · Reviewer_Ffxc · 2025-10-29

**Soundness:** 2
**Presentation:** 2
**Contribution:** 2
**Rating:** 2
**Confidence:** 4

**Summary:**

The paper evaluates whether LLM-generated responses vary when the prompt is in English vs a specific language. They evaluate if prompting in a specific language generates a more culturally appropriate response than a socio-demographic prompting setup in English. They call this gap the explicit-implicit performance gap. Next, they attempt to steer the responses using vectors, which they find are better at reducing culture-specific stereotypes.

**Strengths:**

The paper asks an interesting question about the impact of language in implicitly conditioning open-ended generation. Furthermore, they explore steering vectors as a way to generate localized responses, which also minimize stereotyping.

**Weaknesses:**

1. The paper hypothesizes that language should implicitly indicate a preference for localized response. Hence, the same task, if prompted in a socio-demographic setup using English, should yield semantically similar output if prompted in a regional language without the explicit socio-demographic setup. The assumption that language should succinctly inform the model about the socio-demographic setup might be problematic. This is reflected in the results, which indicate that models possibly maintain a consistent semantic space and need explicit prompting (due to instruction tuning/alignment) to get culture-specific responses. Also, the paper makes a 1:1 language-country assumption by mapping language to a single country, which might be untrue. The example in lines 95-96 might not hold for Bengali spoken in India, where the name "Mohammed" is highly unlikely.
2. The paper makes claims about model stereotyping and diversity ("Despite the explicit prompting benefit, ... toward stereotypes," in the abstract, Table 1). However, (i) they do not formally define stereotyping, and (ii) they only measure stereotyping using LLM as a judge. A formal definition of what entails a generation to be classified as a stereotype should be provided. Human evaluations on a subset of generations should be conducted to enhance the reliability of the LLM-based evaluations. Also, a statistical significance test should be performed on Table 1 to check if the differences are significant. The prompt used with LLMs (Figures 11 and 12) does not contain an option if none of the generated excerpts are stereotypical. Also, would we see the same scores if instead of using a comparative setup, where two excerpts are compared against each other for stereotypicality, each excerpts are scored individually and then their aggregated scores are compared?
3. Section 2.4 defines the EI gap as a difference in probability distributions. What does the subtraction of probability distributions mean in the formula? How do the authors measure distributional difference as a single metric in Figure 1?
4. The paper is very difficult to read, as the ideas and experiments could be conveyed in simpler terms. For example, Sections 3.2 and 3.3 are difficult to read and lack clarity. The authors talk about "option labels" in line 269, without explaining which options they are referring to. I think the issue is that the paper tries to do a lot many things without succinctly studying each thing.  Also, I was consistently getting lost between the jargon - They introduce new terms, such as "non loc prob", "loc prob", etc, without mentioning how they tie to the earlier part of the paper. Figure 3: The formatting of the images should be consistent. The font size and y-axis range should be kept consistent to enhance readability.
5. The paper claims representational universality in lines 312-314, which must be validated statistically. Solely observing the graphs side-by-side does not entail such conclusions.

**Questions:**

See weaknesses

---

> ### Author Response · Authors · 2025-11-24
> **Response to reviewer**
>
> We want to thank the reviewer for their in-depth review and feedback. We hope that we address the concerns of the reviewer below.
>
> W1: The assumption that language should succinctly inform the model about the socio-demographic setup might be problematic. This is reflected in the results, which indicate that models possibly maintain a consistent semantic space and need explicit prompting (due to instruction tuning/alignment) to get culture-specific responses.
>
> R1: Thank you for this note. You actually highlight one of the inspirations of the paper. Past work [1,2] has shown that language models have universal representations of concepts. That being said, for the most part, past work hasn’t studied the impacts of this. For the most part, the impacts have been more anecdotal (like the inspiration in [2] highlighting the Estonian journalist). The point of this paper was to explicitly evaluate real-world situations in which this universal representation might be problematic – prompting a language model only in the language.
>
> [1] Dumas, C., Wendler, C., Veselovsky, V., Monea, G., & West, R. (2025, July). Separating tongue from thought: Activation patching reveals language-agnostic concept representations in transformers. In Proceedings of the 63rd Annual Meeting of the Association for Computational Linguistics (Volume 1: Long Papers) (pp. 31822-31841).
>
> [2] Wendler, C., Veselovsky, V., Monea, G., & West, R. (2024, August). Do llamas work in english? on the latent language of multilingual transformers. In Proceedings of the 62nd Annual Meeting of the Association for Computational Linguistics (Volume 1: Long Papers) (pp. 15366-15394).
>
> W2: Also, the paper makes a 1:1 language-country assumption by mapping language to a single country, which might be untrue. The example in lines 95-96 might not hold for Bengali spoken in India, where the name "Mohammed" is highly unlikely.
>
> R2: Thank you for raising this important point. We agree that linguistic communities are not tied to national borders, and Bengali is a clear example of a language spoken across multiple cultural contexts. Our benchmark does not require that a language corresponds to a single country; rather, our evaluation relies only on relative frequency differences between the culturally local option and the U.S. option within the dominant cultural distribution associated with that language. In the Bengali case, this requires only that the local name be more likely than the American option—not that it be universally common among all Bengali speakers. We recognize that this nuance was not sufficiently clear in the manuscript, and we now clarify this assumption and explicitly incorporate it into the limitations section.
>
> We also note that we’ve added another analysis. In the explicit setting before we added the prefix “I am Bengali”. But now we adjusted the explicit setting to also have “I speak Bengali”. We observe that in this case the explicit-implicit localization gap is the same.
>
> | model             | setting       |     mean | 95% CI         |
> |:------------------|:--------------|---------:|:---------------|
> | Aya Expanse 8B    | explicit      | 0.76259  | [0.733, 0.793] |
> | Aya Expanse 8B    | implicit      | 0.696932 | [0.681, 0.713] |
> | Aya Expanse 8B    | language hint | 0.83369  | [0.822, 0.846] |
> | Gemma 2 27B       | explicit      | 0.878004 | [0.859, 0.896] |
> | Gemma 2 27B       | implicit      | 0.72666  | [0.714, 0.739] |
> | Gemma 2 27B       | language hint | 0.881456 | [0.873, 0.891] |
> | Gemma 2 9B        | explicit      | 0.881356 | [0.862, 0.899] |
> | Gemma 2 9B        | implicit      | 0.603684 | [0.588, 0.620] |
> | Gemma 2 9B        | language hint | 0.838022 | [0.827, 0.848] |
> | Llama 3.1 70B it  | explicit      | 0.879713 | [0.859, 0.898] |
> | Llama 3.1 70B it  | implicit      | 0.739053 | [0.721, 0.758] |
> | Llama 3.1 70B it  | language hint | 0.871773 | [0.862, 0.881] |
> | Llama 3.1 8B base | explicit      | 0.849421 | [0.829, 0.871] |
> | Llama 3.1 8B base | implicit      | 0.543255 | [0.528, 0.560] |
> | Llama 3.1 8B base | language hint | 0.605436 | [0.592, 0.619] |
> | Llama 3.1 8B it   | explicit      | 0.898488 | [0.879, 0.917] |
> | Llama 3.1 8B it   | implicit      | 0.554557 | [0.530, 0.580] |
> | Llama 3.1 8B it   | language hint | 0.64545  | [0.633, 0.661] |
> | gpt-4o            | explicit      | 0.891791 | [0.873, 0.910] |
> | gpt-4o            | implicit      | 0.775908 | [0.758, 0.793] |
> | gpt-4o            | language hint | 0.885327 | [0.877, 0.894] |

---

> ### Author Response · Authors · 2025-11-24
> **Response to reviewer**
>
> W3: (i) they do not formally define stereotyping
>
> R3:We agree that our initial submission did not include an explicit definition. We now provide a formal definition of stereotypicality grounded in prior work defining a generation as stereotypical when it preferentially reflects culturally reductive generalizations rather than diverse or contextually nuanced cultural attributes. This definition is now included in the paper and we add it here for your convenience:
>
> Definition: Stereotypical means that a text relies on oversimplified, fixed beliefs or clichés about a culture. It reduces the complexity of a cultural group to a small number of traits.
>
> W4: (ii) Reliance on LLM-as-a-judge for stereotyping. The prompt used with LLMs (Figures 11 and 12) does not contain an option if none of the generated excerpts are stereotypical.
>
> R4: We appreciate the concern regarding using an LLM evaluator without human validation. We have adjusted the way that we use LLM-as-a-judge for stereotypicality. First, we include a reasoning model (GPT OSS 120B) to add another measure of stereotypicality. We include a table here showing the explicit win-rate for both GPT-OSS and GPT-4o. We note that there is consistency across both models.
>
> Thank you for this. First, we now add a “neither generation is stereotypical” option to the comparison. We find that the times when the model categorized a generation as being equally stereotypical, it now classifies as neither. Although the results are nearly identical across models.
>
> | Model | gpt-4o | gpt-oss-120b |
> |:------|:-------|:-------------|
> | gemma-2-9b | 0.8863 | 0.8586 |
> | gemma-2-27b | 0.8714 | 0.8589 |
> | gpt-4o | 0.8711 | 0.7877 |
> | llama-3.1-70b-instruct-turbo | 0.7958 | 0.8557 |
> | llama-3.1-8b-instruct-turbo | 0.8657 | 0.8575 |
>
> W5: Also, would we see the same scores if instead of using a comparative setup, where two excerpts are compared against each other for stereotypicality, each excerpt is scored individually and then their aggregated scores are compared?
>
> R5: We also re-ran the analysis with where the excerpts are scored individually. We include the results here. We note that the results are the directionally the same as in the win-rate case.
>
> | model                        | metric           |   explicit |   implicit |
> |:-----------------------------|:-----------------|-----------:|-----------:|
> | Gemma_2_27B                  | faithfulness     |    2.83333 |    2.39583 |
> | Gemma_2_9b                   | faithfulness     |    2.53125 |    2.20486 |
> | gpt-4o                       | faithfulness     |    2.93056 |    2.69097 |
> | llama-3.1-70b-instruct-turbo | faithfulness     |    2.33333 |    2.11806 |
> | llama-3.1-8b-instruct-turbo  | faithfulness     |    1.80208 |    1.70833 |
> | Gemma_2_27B                  | stereotypicality |    1.33681 |    1.13194 |
> | Gemma_2_9b                   | stereotypicality |    1.41667 |    1.10764 |
> | gpt-4o                       | stereotypicality |    1.1875  |    1.08333 |
> | llama-3.1-70b-instruct-turbo | stereotypicality |    1.26042 |    1.09028 |
> | llama-3.1-8b-instruct-turbo  | stereotypicality |    1.46875 |    1.26389 |
>
> W6: Section 2.4 defines the EI gap as a difference in probability distributions. What does the subtraction of probability distributions mean in the formula? How do the authors measure distributional difference as a single metric in Figure 1?
>
> R6: Thanks so much for pointing this out. Indeed you're right. The formula was unnecessarily confusing and not aligned with what actually did. We updated the formula and the text. The update is being included here:
>
> $$\text{EI-Gap} = \mathbb{E}[\mathbf{1}\{\hat{y}(x_{\text{context}} \circ x_q)=y\}]-\mathbb{E}[\mathbf{1}\{\hat{y}(x_{\text{tr}q})=y_{\text{tr}}\}].$$
>
> In which $\mathbb{E}$ denotes an expectation over dataset items, $\mathbf{1}\{\cdot\}$ is an indicator for whether the model’s prediction exactly matches the correct culturally localized answer, $\hat{y}(\cdot)$ is the model’s produced output, $x_{\text{context}} \circ x_q$ is the explicitly localized prompt, and $x_{\text{tr.}q}$ is the implicitly localized translated prompt. The first term reflects average correctness under explicit localization, while the second term reflects average correctness under implicit localization.

---

> ### Author Response · Authors · 2025-11-24
> **Response to reviewer**
>
> W7: The paper is very difficult to read, as the ideas and experiments could be conveyed in simpler terms. For example, Sections 3.2 and 3.3 are difficult to read and lack clarity. The authors talk about "option labels" in line 269, without explaining which options they are referring to.
>
> R7: When modifying the options, we separate the tokens that come from the source prompt and the target prompt. This helps us identify the layers at which the source prompt starts to dominate the model’s behavior (as shown with the green line at Figure 3).
>
> W8: They introduce new terms, such as "non loc prob", "loc prob", etc, without mentioning how they tie to the earlier part of the paper. Figure 3.
>
> R8: Thank you for noting this. We have now mentioned non loc prob and loc prob in the main paper.
>
> W9: The paper claims representational universality in lines 312-314, which must be validated statistically. Solely observing the graphs side-by-side does not entail such conclusions.
>
> R9: Thank you for raising this. We realize our wording was misleading. We are not claiming any formal notion of “representational universality.” Rather, we empirically observe that both explicit and implicit localization signals tend to peak in the middle layers, and we report confidence intervals for these probabilities. We will revise the text to avoid the term “universality” and make clear that we are not asserting that these behaviors are statistically identical.

---

> > ### Comment · Reviewer_Ffxc · 2025-11-24
> >
> > I thank the reviewer for their response. I think the paper would benefit if the motivation could be strengthened and the experiments statistically validated.
> >
> > 1. If I understand correctly, the study is proposed as an LLM bias study to measure their suitability across 5 different cultures. However, I find the underlying motivation somewhat problematic. Since models generate text through maximum likelihood estimation (MLE) decoding, this process collapses a rich probability distribution into a single output. As a result, if one culture perceives the color of a pumpkin as green and another as orange, the model's single response will inevitably be "incorrect" for one of them. This does not necessarily indicate that the model lacks cultural competence. It simply reflects the nature of deterministic decoding. In such cases, context plays a crucial role. Socio-demographic prompting, where the cultural context is explicitly specified, is essential for producing culturally grounded outputs. Thus, expecting a single, context-free answer from the model risks enforcing a stereotype rather than revealing genuine cultural sensitivity, which the framework attempts. Hence, a more appropriate approach would be to evaluate the distribution of responses rather than individual generations. For instance, sampling the model's outputs over K trials using different temperature settings could reveal how the model's probability mass is distributed across culturally variant options. Such a setup would more directly address the authors' hypothesis about model bias. Finally, I would argue that the results presented in Figure 1 and Figure 2 reflect properties of the prompts more than of the model itself. The observed variation in responses with different prefixes suggests that the prompt framing, rather than the model's internal bias, largely determines the outcome. So, I am not sure if the findings reflect the model's incapabilties. On the contrary, they suggest that models avoid stereotyping by remaining sensitive to explicit contextual cues rather than implicitly encoding cultural assumptions.
> >
> > 2. Although using LLM-as-judge is fine, it should not be the gold standard, especially while evaluating stereotypes. Hence, measuring alignment of the LLM-as-judge scores with humans would be prudent. Also, all the reported numbers and their differences must be statistically validated.
> >
> > 3. The writing still sounds dense and missing explanations where needed. For example, the dataset samples should be in the main body rather than the appendix. The mechanistic interpretability study seems like a side quest and is not adequately motivated. Hence, it might be placed in the appendix, with more focus on the bias measurements, their formalisms, and statistical tests.
> >
> > I thank the authors and reserve my current scores.

---

> > > ### Author Response · Authors · 2025-11-29
> > >
> > > We want to thank the reviewer for their comments.
> > >
> > > > Since models generate text through maximum likelihood estimation (MLE) decoding, this process collapses a rich probability distribution into a single output. As a result, if one culture perceives the color of a pumpkin as green and another as orange, the model's single response will inevitably be "incorrect" for one of them. This does not necessarily indicate that the model lacks cultural competence.
> > >
> > > - We would like to push back a little on this assessment. This assumes that we are prompting the language model in one context; however, like we describe in the introduction if we ask the model in Japanese what color is a pumpkin we would want it to answer in a way that is aligned with what would be expected from someone from Japan. The MLE should differ depending on the language you prompt the model in.
> > >
> > > >  In such cases, context plays a crucial role.
> > >
> > > - We agree and argue that language should be sufficient context.
> > >
> > > > Socio-demographic prompting, where the cultural context is explicitly specified, is essential for producing culturally grounded outputs.
> > >
> > > - We see your point, but in our paper we also note downsides with explicit prompting like increased homogeneity and stereotypically.
> > >
> > > > Hence, a more appropriate approach would be to evaluate the distribution of responses rather than individual generations. For instance, sampling the model's outputs over K trials using different temperature settings could reveal how the model's probability mass is distributed across culturally variant options. Such a setup would more directly address the authors' hypothesis about model bias.
> > >
> > > - We agree with the reviewer that an interesting analysis would be to compare either K trials or looking directly at the logits. Nevertheless, we want to note to the reviewer that this is unlikely how individuals from these cultures would use these models. If someone asks a model in Russian "what city am I more likely to live in Toronto or Moscow?" They would expect the model to answer Moscow, but now we find that the model consistently answers Toronto. This is the problem we're exploring.
> > >
> > > > Finally, I would argue that the results presented in Figure 1 and Figure 2 reflect properties of the prompts more than of the model itself. The observed variation in responses with different prefixes suggests that the prompt framing, rather than the model's internal bias, largely determines the outcome. So, I am not sure if the findings reflect the model's incapabilties. On the contrary, they suggest that models avoid stereotyping by remaining sensitive to explicit contextual cues rather than implicitly encoding cultural assumptions.
> > >
> > > - We believe that the reviewer may have a misunderstanding about what Figure 1 and 2 are showing; unless we ourselves are misunderstanding the comment. In figure 1 we show the explicit-implicit localization gap. In figure 2 we adjust the implicit setting to include a random word associated with the culture of the implicit prompt. We find that just by adding a random culturally-salient word is enough to reduce the gap. Thus, reinforcing the argument that language alone doesn't have enough information to culturally localize a generation.
> > >
> > > > Although using LLM-as-judge is fine, it should not be the gold standard, especially while evaluating stereotypes. Hence, measuring alignment of the LLM-as-judge scores with humans would be prudent. Also, all the reported numbers and their differences must be statistically validated.
> > >
> > > - We agree with this point and are currently running a human validation. We'll hopefully have the results in time.
> > >
> > > > The writing still sounds dense and missing explanations where needed. For example, the dataset samples should be in the main body rather than the appendix. The mechanistic interpretability study seems like a side quest and is not adequately motivated. Hence, it might be placed in the appendix, with more focus on the bias measurements, their formalisms, and statistical tests.
> > >
> > > - We thank the reviewer for this point and will add the dataset to the main text.

---

### Official Review · Reviewer_gLzr · 2025-11-04

**Soundness:** 3
**Presentation:** 4
**Contribution:** 3
**Rating:** 8
**Confidence:** 3

**Summary:**

The paper demonstrates the presence of an explicit-implicit cultural localization gap, where the models fail to generate the correct answers in a culture’s language, even when they do when the culture is explicitly mentioned in the prompt. The paper proposes a steering based approach to improve this localization. The results show that for 5 cultures, culture-specific vectors help reduce this gap while also maintaining diversity of answers in open-ended generations (which explicit prompting reduces).

**Strengths:**

- The paper clearly shows a gap and proposes a well grounded solution
- There is thorough analysis of the relevant cultural information within the model, with individual steering vectors as well as combined
- The paper is clearly written and easy to follow

**Weaknesses:**

- 3/4 benchmarks on which the results are shown are new, created as part of this study but not enough information is provided to validate their effectiveness.
- The choice of language/cultures aren't ones that are localised to one culture - for instance Bengali is spoken in both Bangladesh and West Bengal in India, which could explain the difference in performance

**Questions:**

NA

---

> ### Author Response · Authors · 2025-11-24
> **Response to reviewer**
>
> We want to thank the reviewer for taking the time to review the paper and provide their feedback. We answer some of the reviewers questions and concerns below:
>
> W1: 3/4 benchmarks on which the results are shown are new, created as part of this study but not enough information is provided to validate their effectiveness.
>
> R1: Thank you for this note. Out of curiosity, what would effectiveness look like for you? We think it makes sense to give some background into how we created these 3 datasets. Studying cultural localization is tricky because you want model answers that are unique to a specific culture. “What is the force of gravity?” clearly doesn’t work because the answer is culture invariant. We spent a lot of time exploring how to make a good cultural localization dataset or take from other people. There were insufficiencies in a lot of the datasets for this concept. So in the end, we focused on manually making datasets that very likely give different results depending on the culture that’s asking the question. We agree they might not be super “effective” in the usual sense of evals. But for our task of measuring cultural localization we thought it was the right framing.
>
> W2: The choice of language/cultures aren't ones that are localised to one culture - for instance Bengali is spoken in both Bangladesh and West Bengal in India, which could explain the difference in performance
>
> R2: Thank you for pointing this out. We agree that this is indeed a concern, and a tough part of studying multilinguality / culture more generally. However, it’s worth noting that in the explicit setting even when we add a random word like “biriyani” [বিরিয়ানি] then the model is able to correctly localize the generation. Biriyani is a food both in Bangladesh and India. We describe this in Section 3.1 ‘Characterizing explicit localization’.
> We’ve now added another analysis where instead of stating the culture of the person in the explicit setting, we state what language the person speaks -- reconciling the possible explanation you mention. So instead of giving the model information about the culture of the person, we only provide the model with the language that the person speaks. We observe similar patterns here.
>
> In the table below for the models we show the localization by just adding “I speak language X” – denoted as “language hint” in the setting. We find that the language hint itself is enough to localize the generation.
>
> | model             | setting       |     mean | 95% CI         |
> |:------------------|:--------------|---------:|:---------------|
> | Aya Expanse 8B    | explicit      | 0.76259  | [0.733, 0.793] |
> | Aya Expanse 8B    | implicit      | 0.696932 | [0.681, 0.713] |
> | Aya Expanse 8B    | language hint | 0.83369  | [0.822, 0.846] |
> | Gemma 2 27B       | explicit      | 0.878004 | [0.859, 0.896] |
> | Gemma 2 27B       | implicit      | 0.72666  | [0.714, 0.739] |
> | Gemma 2 27B       | language hint | 0.881456 | [0.873, 0.891] |
> | Gemma 2 9B        | explicit      | 0.881356 | [0.862, 0.899] |
> | Gemma 2 9B        | implicit      | 0.603684 | [0.588, 0.620] |
> | Gemma 2 9B        | language hint | 0.838022 | [0.827, 0.848] |
> | Llama 3.1 70B it  | explicit      | 0.879713 | [0.859, 0.898] |
> | Llama 3.1 70B it  | implicit      | 0.739053 | [0.721, 0.758] |
> | Llama 3.1 70B it  | language hint | 0.871773 | [0.862, 0.881] |
> | Llama 3.1 8B base | explicit      | 0.849421 | [0.829, 0.871] |
> | Llama 3.1 8B base | implicit      | 0.543255 | [0.528, 0.560] |
> | Llama 3.1 8B base | language hint | 0.605436 | [0.592, 0.619] |
> | Llama 3.1 8B it   | explicit      | 0.898488 | [0.879, 0.917] |
> | Llama 3.1 8B it   | implicit      | 0.554557 | [0.530, 0.580] |
> | Llama 3.1 8B it   | language hint | 0.64545  | [0.633, 0.661] |
> | gpt-4o            | explicit      | 0.891791 | [0.873, 0.910] |
> | gpt-4o            | implicit      | 0.775908 | [0.758, 0.793] |
> | gpt-4o            | language hint | 0.885327 | [0.877, 0.894] |

---

> > ### Comment · Reviewer_gLzr · 2025-11-24
> >
> > Thank you for the response and providing the additional results!
> > > Out of curiosity, what would effectiveness look like for you?
> >
> > By effectiveness of the benchmark, I mean validity of the prompts/data used for extraction of cultural knowledge. For the o1 distilled part of the benchmark, the data is entirely synthetically generated but no human validation is done to ensure that it is valid. I can't see what the methodology for collection is for the names data source either. The response provided by the authors that certain crowd workers use LLMs anyway is not a valid reason to not conduct robust human validation, in my opinion. There are platforms other than the one that was tested and ways of ensuring limited use of LLMs in annotation.
> > > There were insufficiencies in a lot of the datasets for this concept
> >
> > This is a fair reason, but I think these insufficiencies should be elaborated on in the paper. Here's an informal list of some of existing sources of cultural knowledge benchmarks is present here: https://github.com/simran-khanuja/awesome-cultural-nlp?tab=readme-ov-file#dataset. I don't immediately see why some of those are incompatible with what the authors are trying to show. Strong results on existing data sources only provide more validity to the claims made in the paper.

---

### Author Response · Authors · 2025-11-24
**Thank you to the reviewers**

We want to thank the reviewers for their thoughtful comments and feedback. We believe all of these comments have significantly strengthened the paper. In this note we’ll first summarize our view on the key contributions of the paper and then get into how we addressed the common themes of the reviewers feedback.

Past work has shown that language models have universal concept representations, regardless of the language that you prompt the language in. While this has been shown in toy settings, no papers have explored the real-world impacts of this. If someone prompts a language model out-of-the-box in Bengali, will it answer in a way that’s faithful to that culture.

To study this we operationalize a simple dataset where the answers should differ across cultures. We then measure performance across two settings: directly inputting the culture of the person the language model is generating for and no direct information beyond the language. We compare the difference of having explicit context vs. only implicit context as the explicit-implicit localization gap. We then evaluate if this gap exists in various language models and find that it does.

Is explicit context all we need? No. This is not how real people prompt language models and it leads to more stereotypical and homogenous generations.

To get around this we design a simple steering approach where we add a linear vector to a specific layer in the LLM and show that this leads to improved localization without a reduction in homogeneity / stereotypicality.

Points of the reviewers:
1. Clarity in terms of explanations and definitions.
   - We want to thank the reviewers for addressing where they felt our explanations were unclear. In turn, we adjusted the writing of the manuscript in these places.
   - We define in the main text how we define “stereotypical”.
   - We updated the EI-Gap formula.
   - We fixed the grammatical errors noted by reviewer sbiU

2. Confidence intervals.
   - We’ve now added bootstrapped confidence intervals to the steering results (Table 2 and 4). We note that the confidence intervals are tight and do not change the results.

3. Culture and language.
   - One of the largest critiques by the papers is the claim that language and culture are different and our assessment that language alone should be able to culturally localize a generation is wrong.
   - To adjust for this we ran a new experiment where we tried to explicitly localize the model by adding in the prompt “I speak language X” (instead of “I live in country X”). This gets at the problem where someone from West Bengal may be culturally different from someone from Bangladesh yet speak the same language. We note that in this context the model is able to successfully culturally localize the answers from that country. We added the tables to each of the reviewers that mentioned this point.
   - We think this may be because what we actually use is much weaker: we only need the “local” option to be more common than the U.S. option within the main cultural group tied to that language. For Bengali, that means Mohammed just needs to be more likely than George in the Bangladeshi Bengali distribution—not among all Bengali speakers. We didn’t spell this out clearly, so we’ve now clarified it and added it to the limitations.

4. Concern about using GPT-as-a-judge for measuring faithfulness / stereotypicality / fluency:
   - We didn’t run human evaluations because, in practice, many evaluators end up delegating to LLMs anyway [1], which collapses the signal we’re trying to measure. Still, we took the concern seriously. We added a second model-as-judge to check that the results don’t depend on GPT alone and now reasons – the patterns hold. We also addressed the request for a “neither–stereotypical” option. Together, these give a more stable and model-agnostic read on stereotypicality and faithfulness.

5. Inadequate exploration of effects of steering:
   - We note that the steering results are not the main contribution of the paper, but we do agree that its implications should have been better studied. We now add to the Appendix B3 results of measuring fluency of the steered vs. non-steered generations. We found that steering does not reduce fluency.

[1] Veniamin Veselovsky, Manoel Horta Ribeiro, Philip J. Cozzolino, Andrew Gordon, David Rothschild, and Robert West. 2025.
Prevalence and Prevention of Large Language Model Use in Crowd Work. Commun. ACM 68, 3 (March 2025), 42–47.

---

### Note · Authors · 2026-01-05

**Comment:**

Thank you to all the reviewers for their feedback. We've decided to withdraw the paper.

**Withdrawal Confirmation:**

I have read and agree with the venue's withdrawal policy on behalf of myself and my co-authors.